# Oxygen export to the deep ocean following Labrador Sea Water formation

Jannes Koelling[1], Dariia Atamanchuk[1], Johannes Karstensen[2], Patricia Handmann[2], and Douglas W.R. Wallace[1]

[1]Dalhousie University, Halifax, Nova Scotia, Canada
[2]GEOMAR Helmholtz Centre for Ocean Research, Kiel, Germany

**Correspondence:** Jannes Koelling (j.koelling@dal.ca)

**Abstract.** The Labrador Sea in the North Atlantic Ocean is one of the few regions globally where oxygen from the atmosphere can reach the deep ocean directly. This is the result of wintertime deep convection, which homogenizes the water column to a depth of up to 2000 m, and brings deep water undersaturated in oxygen into contact with the atmosphere. In this study, we analyze how the intense oxygen uptake during Labrador Sea Water (LSW) formation affects the properties of the outflowing deep western boundary current, which ultimately feeds the upper part of the North Atlantic Deep Water layer in much of the Atlantic Ocean.

Seasonal cycles of oxygen concentration, temperature, and salinity from a two-year time series collected by sensors moored at 600 m nominal depth in the outflowing boundary current at 53° N show a cooling, freshening, and increase in oxygen content of the water flowing out of the basin between March and August. Analysis of Argo float data suggests that this is preceded by an increased input of LSW into the boundary current about one month earlier. This input is the result of newly ventilated LSW entering from the interior as well as LSW formed directly within the boundary current. Together, these results imply that the southward export of newly formed LSW primarily occurs in the months following the onset of deep convection, from March to August, and that this direct LSW export route controls the seasonal oxygen increase of the outflow at 600 m depth. During the rest of the year, properties of the boundary current measured at 53° N resemble those of Irminger Water, which enters the basin with the boundary current from the Irminger Sea.

The input of newly ventilated LSW increases the oxygen concentration from 298 $\mu$mol L$^{-1}$ in January to a maximum of 306 $\mu$mol L$^{-1}$ in April. As a result of this LSW input, an estimated $(1.60 \pm 0.42) \times 10^{12}$ mol year$^{-1}$ of oxygen are added to the outflowing boundary current, mostly during spring and summer, equivalent to 50% of the wintertime uptake from the atmosphere in the interior of the basin. The export of oxygen from the subpolar gyre associated with this direct southward pathway of LSW is estimated to supply 42–71% of the oxygen consumed annually in the upper North Atlantic Deep Water layer in the Atlantic Ocean between the equator and 50° N. Our results show that the formation of LSW is important for replenishing oxygen to the deep oceans, meaning that possible changes in its formation rate and ventilation due to climate change could have wide-reaching impacts on marine life.

# 1 Introduction

Much of the global supply of oxygen to the deep ocean is concentrated in a few key regions where near-surface water sinks to great depth and spreads away from its source region (Talley, 2008; Gebbie and Huybers, 2011; Khatiwala et al., 2012). This process ventilates the deep ocean, supplying oxygen to a vast volume of water that would otherwise be barren, and making it capable of sustaining life (Rogers, 2015; Isozaki, 1997). One of the regions where such deep ocean ventilation occurs is the Labrador Sea, a semi-enclosed marginal sea nestled between eastern Canada and western Greenland. Here, strong

wintertime atmospheric cooling leads to buoyancy loss of surface waters that lay above a weakly stratified water column, causing convective overturning that homogenizes the upper 1000-2000 m of the ocean (Marshall and Schott, 1999; Yashayaev, 2007). Due to the great depths reached by convection in this region, it is generally referred to as deep convection in order to differentiate it from the much shallower-reaching convection occurring in mixed layers throughout the global ocean, but we will use the terms convection and deep convection interchangeably hereinafter in the interest of brevity. During the convection

season, which typically lasts from January to April, deep water masses that are low in oxygen are continuously incorporated into the progressively deepening mixed layer, which leads to a decrease in near-surface oxygen. This results in severe air-sea gradients in oxygen concentration, which, together with extreme atmospheric conditions, drive intense uptake of oxygen during winter (Koelling et al., 2017; Wolf et al., 2018; Atamanchuk et al., 2020).

The climatological flow field in the Labrador Sea features a cyclonic boundary current entering at the southern tip of Green-

land and exiting at the southwestern end of the Labrador Sea, with only a weak mean flow in the interior (Fig. 1). At mid-depth, the boundary current entering the basin carries Irminger Water (IW), a water mass originating in the Atlantic ocean and modified in the subpolar gyre, particularly the Irminger Sea (Cuny et al., 2002; Pacini et al., 2020). The cyclonic circulation in the Labrador Sea is linked to a doming of isopycnals in the center of the basin, which supports deep convection in the interior (Marshall and Schott, 1999). Different types of eddies, such as Irminger Rings and convective eddies, compensate the annual

mean surface heat loss in the interior, and help restratify the water column above the water mass formed during convection, known as Labrador Sea Water (LSW) (Eden and Böning, 2002; Straneo, 2006; de Jong et al., 2014; Rieck et al., 2019). Variability in atmospheric forcing, lateral fluxes of heat and freshwater, and the evolution of the mixed patch are main drivers for interannual variability in the depth of convection and in properties of LSW (Lazier, 1973; Yashayaev and Loder, 2016). Much of the research has been focused on the deep convection region in the interior of the basin, but it has been shown that convection

can also occur within the boundary current itself (Pickart et al., 1997; Palter et al., 2008). The densest water masses formed in the boundary current are similar to the "classical" LSW formed in the interior, while convection over the continental slope forms a lighter water mass known as upper Labrador Sea Water, or uLSW (Pickart et al., 2002; Cuny et al., 2005). While the relative importance of boundary current convection is still uncertain, some model studies have suggested that it could account for a substantial fraction of the LSW that is eventually exported out of the region (Brandt et al., 2007; MacGilchrist et al.,

2020).

The LSW formed each winter in the center of the basin is distinguishable in summertime hydrographic sections as a cold, fresh water mass with high oxygen content, compared to the warmer, saltier IW found in the boundary current near Greenland

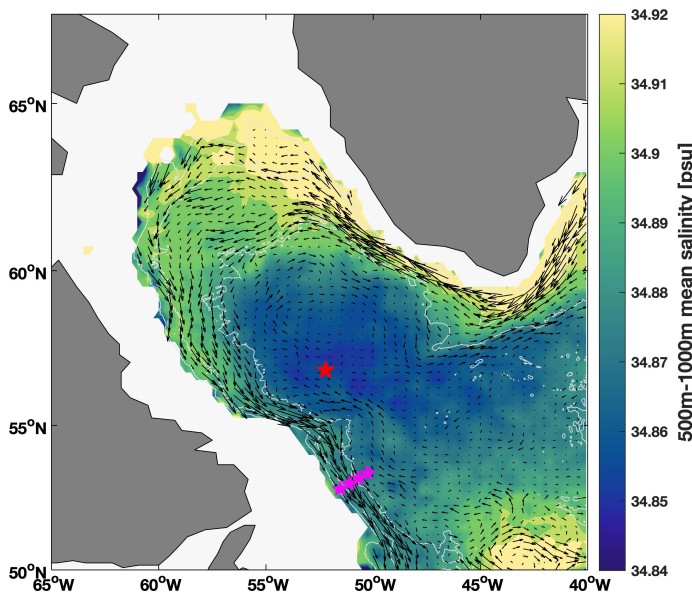

**Figure 1.** Mean salinity between 500 m and 1000 m in the Labrador Sea, calculated from all available Argo float profiles between 2000 and 2020 (see Sec. 2.2 for details), averaged within overlapping $0.25° \times 0.25°$ bins. Gray lines show the 1000 m, 2000 m and 3000 m isobaths, vectors show mean currents at the depth of LSW from Fischer et al. (2018), and symbols correspond to positions of the SeaCycler mooring (red) and the 53° N array (magenta).

(see Yashayaev and Loder (2016) and Zou et al. (2020) for sections across the Labrador Sea). LSW is one of the main water masses making up North Atlantic Deep Water, (NADW) (Lazier, 1973), and spreads throughout the Atlantic Ocean, both as part of the Deep Western Boundary Current (Molinari et al., 1998; Toole et al., 2017) and in the interior of the basin (Bower et al., 2009; Lozier, 2010). The signature of LSW is distinguishable in NADW properties far from the source region (Talley and McCartney, 1982; Rhein et al., 2004; Le Bras et al., 2017), including elevated oxygen concentrations compared to adjacent water masses (Atkinson et al., 2012). Although the importance of ventilation in the Labrador Sea on NADW properties is readily apparent from hydrographic data, the exact timing and mechanisms of the southward export of LSW from the subpolar gyre are not well established. Due to the weak mean currents in the center of the basin (see Fig. 1), some studies have suggested that much of the newly formed LSW remains in the interior for several years, with only a small fraction exported in the months following convection (Rhein et al., 2002; Straneo et al., 2003; MacGilchrist et al., 2021). On the other hand, there is evidence of a rapid export pathway of newly formed LSW, which may be associated with convection either within or close to the boundary current (Pickart et al., 1997; Brandt et al., 2007). An analysis from float data by Palter et al. (2008) showed that both interior and boundary convection can affect the properties of the outflowing boundary current, and suggested that eddies may play an important role in the input of newly formed LSW into the boundary current.

In this study, we use a novel data set including moored oxygen concentration measurements from the outflowing boundary current at the southern exit of the Labrador Sea. The data are recorded at the 53° N array (Zantopp et al., 2017), which is part of the Overturning in the Subpolar North Atlantic Program, OSNAP (Lozier et al., 2017). We analyze annual cycles of oxygen, temperature, and salinity, along with data from Argo floats, in order to investigate seasonal variations in the input of LSW into the boundary current and its export out of the basin. Our results highlight how changes in properties of the outflowing boundary current over a seasonal cycle, and resulting changes in the export of oxygen, are linked to the formation and export of newly ventilated LSW.

## 2 Data and Methods

### 2.1 Moored sensor data

Data used in this study were collected from May 2016 to May 2018 on the 53° N array in the boundary current at the exit of the Labrador Sea, on moorings K7 (52.86° N, 51.31° W), K8 (52.96° N, 51.31° W), K9 (53.14° N, 50.87° W), and K10 (53.39° N, 50.25° W) (west to east in Fig. 1). Moorings have been deployed at the location since 1997, but in various configurations (Zantopp et al., 2017). In the present setup, each mooring is equipped with Aquadopp and RCM current meters and SeaBird SBE 37 instruments measuring conductivity, temperature (T), and pressure, which were also used to derive salinity (S). The depths and positions of the instruments are optimized to measure the strength and properties of the boundary current exiting the Labrador Sea, including transports of different layers (Zantopp et al., 2017). Since 2016, Aanderaa 4330O optodes (Aanderaa Data Instruments, AS) have been deployed at select depths alongside T/S sensors to measure dissolved oxygen ($O_2$) concentrations (Fig. 2a). We also use data from an Aanderaa 4330 oxygen optode mounted at 500 m depth on the SeaCycler mooring in the central Labrador Sea (see Fig. 1 for location). For the Seacycler optode, no co-located salinity data are available, and we estimate salinity used in Sec. 3.2 from a climatological T/S-relationship at 500 m calculated from Argo float data.

The deployment depths of the oxygen sensors on the 53° N array are shown in Fig. 2 along with mean oxygen and salinity sections along the array recorded with a Conductivity-Temperature-Depth (CTD) rosette. The majority of the $O_2$ sensors were deployed at nominal depths of approximately 600 m or 1900 m, chosen to represent the LSW core, and a depth near the bottom of the LSW layer. Only the sensors at 600 m depth are used in this study, and the deployment and calibration information for these sensors is summarized in Table 1. LSW is commonly defined in density space by a range of potential density $\sigma_\theta$, with a lower boundary of 27.8 $\mathrm{kg\,m^{-3}}$, and an upper boundary of either 27.68 $\mathrm{kg\,m^{-3}}$ (Pickart et al., 1997; Zantopp et al., 2017; Lozier et al., 2019) or 27.7 $\mathrm{kg\,m^{-3}}$ (Zou et al., 2020), the former of which is used as the definition of the LSW layer in this study. Within the LSW layer, oxygen concentrations are elevated above 1200 m, and the mean salinity is 34.86. The low-oxygen core below LSW centered at about 2000 m depth coincides with a salinity maximum, which is indicative of Northeast Atlantic Deep Water (NEADW) (Yashayaev, 2007). Some authors differentiate between "upper" and "classical" LSW (Pickart et al., 1997), with the boundary between the two water masses typically at a potential density of $\sigma_\theta = 27.74$ $\mathrm{kg\,m^{-3}}$ (Kieke et al., 2006). The mean potential density at 600 m depth is 27.72 $\mathrm{kg\,m^{-3}}$ for K7, and 27.74 $\mathrm{kg\,m^{-3}}$ for K8–K10 (Table 1), suggesting that sensors on these three moorings are near the interface between uLSW and cLSW (see also Fig. 2).

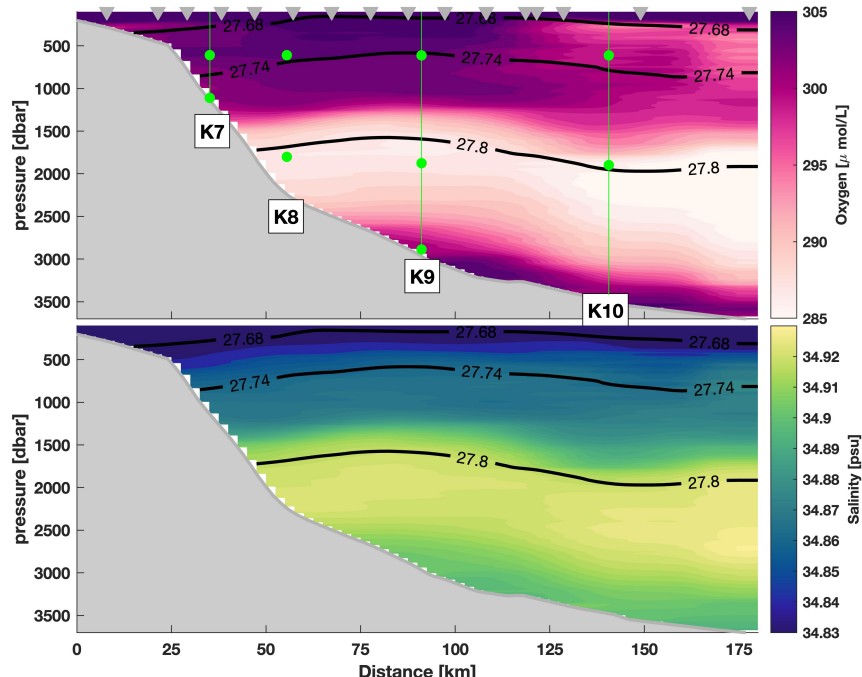

**Figure 2.** Oxygen and salinity sections along the array from shipboard measurements collected during four mooring deployment and recovery cruises (see Sec. 2.2). Gray symbols show stations from one representative cruise. Green symbols in the top panel show deployment locations of the oxygen sensors, which were co-located with T/S sensors. Only data from the instruments at 600 m depth are used in this study. The full coverage of current meters and T/S sensors is described in Zantopp et al. (2017). Black contours show the potential density range of $27.68 \, \mathrm{kg \, m^{-3}} \leq \sigma_\theta \leq 27.8 \, \mathrm{kg \, m^{-3}}$ for LSW, and the $\sigma_\theta = 27.8 \, \mathrm{kg \, m^{-3}}$ isopycnal which delineates the upper Labrador Sea Water (uLSW) above and classical Labrador Sea Water (cLSW) below.

The mean salinity of the boundary current in the LSW depth range decreases as it flows cyclonically around the basin (Fig. 1). At the 53° N array, the salinity is closer to values found in the interior Labrador Sea compared to the Greenland side, consistent with the view that there is significant input of LSW from the interior. However, the water here is still warmer and saltier than that found in the convective interior, suggesting that it is a mixture of LSW and IW, which enters the Labrador Sea on the northern side in a mean core depth of about 500 m, and can be seen as a local salinity maximum near this depth (Pacini

et al., 2020). In this study, we will focus on the variability of properties in this LSW layer, using data from the sensors mounted at about 600 m on the moorings, representing the full width of the boundary current from the shelf break area (K7, K8), via the core (K9), into the outer edge and recirculation regime (K10).

### 2.1.1   Sensor calibration

Temperature and salinity sensors were calibrated following the procedure outlined in Karstensen (2005). Briefly, the method

involves attaching the sensors to a CTD rosette prior to and after each deployment, and comparing the data from the different

| Mooring | Nominal deployment depth [m] | Mean potential density [kg m$^{-3}$] | Optode Serial number | Drift [µmol L$^{-1}$] |
|---------|------------------------------|--------------------------------------|----------------------|------------------------|
| K7 | 608 | 27.72 | 052634 | -5.8 |
| K8 | 608 | 27.74 | 052631 | -5.2 |
| K9 | 610 | 27.74 | 052628 | -2.5 |
| K10 | 610 | 27.74 | 052626 | -6.1 |

**Table 1.** Deployment locations, depths and drifts of the oxygen optodes during 2016–2018.

instruments at certain stop depths with the CTD recording, which is calibrated with discrete bottle samples taken throughout the cruise.

Oxygen optodes were supplied with individual multipoint factory calibration, with an absolute accuracy of 1.5% or 2 µmol L$^{-1}$ (Tengberg et al., 2006; Tengberg and Hovdenes, 2014), and powered by single-channel loggers (RBR Ltd.). The optodes were further calibrated against Winkler samples collected during CTD casts at the mooring locations during deployment and recovery cruises in 2016 and 2018 to correct for sensor drift. The drift of the sensors during deployment ranged from -2.5 µmol L$^{-1}$ to -6.2 µmol L$^{-1}$, or 0.4–1% per year (see Table 1).

## 2.2 Additional data sets

In addition to the mooring data that are the focus of this study, we use ancillary data from a number of different sources, which will be briefly described in this section

CTD data used in Fig. 2 were collected during four mooring deployment cruises in 2014 (cruise Thalassa MSM40), 2016 (MSM54), 2018 (MSM74), and 2020 (MSM94). The station spacing near the moorings was typically around 10 km, and the stations occupied during the 2016 cruise are indicated in the figure. The sections shown in the figure were produced by interpolating all measurements from the four cruises onto a regular grid with 2.5 km horizontal and 1 dbar vertical spacing using a gaussian interpolation.

We also use data from the Argo program (Roemmich et al., 2009), which are freely available online, including both individual float data and data products. The data from individual floats were downloaded from coriolis.eu.org, using all profiles flagged as "good" on the data selection service that were found in the area of 65°W–40°W, 50°N–65°N between Jan 1, 2000 and Sep 8, 2020 (Argo (2020), https://doi.org/10.17882/42182#77634), resulting in a total of 41165 temperature and salinity profiles from 568 floats. These data were used to produce the mean salinity map in Fig. 1, and are used to track the formation and boundary current input of LSW as described in Sec. 2.3. Data from the Argo mixed layer depth database by Holte et al. (2017) are used in Fig. 7 in order to highlight the mean convection area.

Bottom topography shown in Figs. 1, 3a, and 7, comes from the SRTM15+ product (Tozer et al., 2019), which is the most recent product based on Smith and Sandwell (1997). These data are also used to differentiate between the interior and boundary current regions for the analysis of Argo data. Current vectors used in Fig. 1 are taken from the dataset described in Fischer et al. (2018).

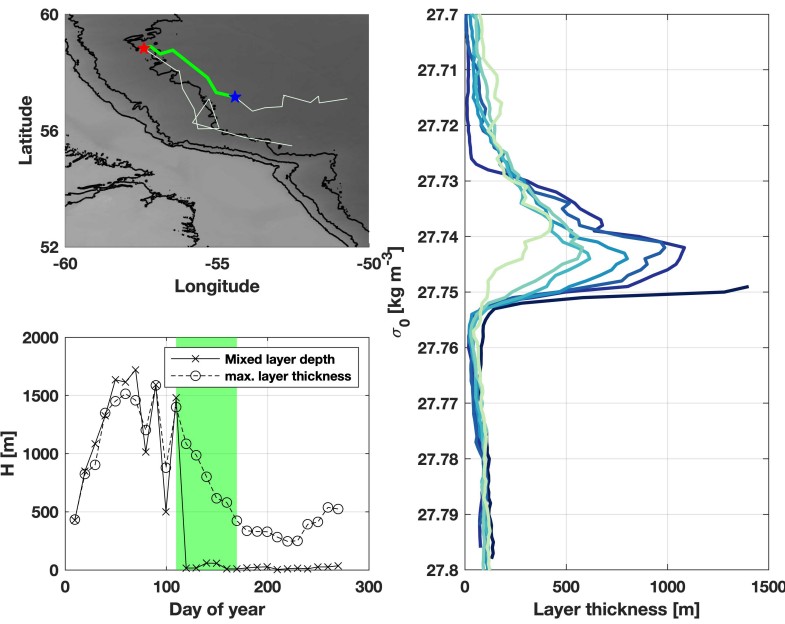

**Figure 3.** Example of Argo data used to track LSW input into the boundary current after convection, from float ID 6902589. a) Trajectory showing latest deep mixed layer measurement (blue star), location of entry into boundary current (red star). Thick green line highlights the part of the trajectory from LSW formation to crossing the 3000 m isobath. b) Time series of mixed layer depth and maximum layer thickness, with green shading highlighting the time from convection in the interior to entering the boundary current. c) Consecutive profiles of layer thickness in density space, going from convection (dark blue) to input into boundary current (light green).

## 2.3 Tracking Labrador Sea Water formation and input into the boundary current

To evaluate the input of LSW into the outflowing boundary current, we use data from Argo floats to investigate convection regions and LSW input in Sec. 3.1 and 3.2. The profile data are used to calculate mixed layer depths based on a density difference of $\Delta\sigma_\theta = 0.01 \ \mathrm{kg\,m^{-3}}$ from the shallowest measurement, which has been shown to be a suitable threshold in the subpolar North Atlantic (Piron et al., 2016; Zunino et al., 2020). After the last convective profile for each season, defined here as a mixed layer depth exceeding 600 m, we track the trajectory until the float enters the boundary current. We define LSW input as a float crossing the 3000 m isobath and subsequently staying in the boundary current for at least 2 out of the next 3 profiles thereafter, similar to definitions used in previous studies (Georgiou et al., 2020). A stricter criterion was also tested, requiring floats to subsequently be exported south of 53°N within the boundary current, reducing the number of floats used for the analysis by about half. However, results are not sensitive to this choice, with the curve shown in Fig. 9b being almost identical if this export criterion is used instead.

During a typical Argo cycle, floats park at a depth of either 1000 m or 1500 m for 10 days before descending to 2000 m, measuring a profile to the surface and staying there to transmit the data, which typically last about one hour, but can be as much as 12 hours for older float models (Lebedev et al., 2007). As a result, the floats are not strictly following a water parcel, as vertically sheared currents can carry them away from the water mass they had been advected with at depth. To ensure that floats used for the calculation are following LSW, we use layer thickness measurements to track them. Layer thickness is inversely proportional to the vertical density gradient, and newly formed LSW is readily detected as a maximum (Yashayaev and Loder, 2016). An example of layer thickness measurements from a float measuring convection in the interior of the basin and subsequently entering the boundary current on the southwestern side is shown in Fig. 3. Initially, the time series show mixed layer depth and maximum layer thickness varying in concert, as the thickest layer is the actively convecting mixed layer (Fig. 3b). This is also evident in the last profile measuring convection, as the maximum layer thickness is found at the lowest density measured, suggesting that no lighter water mass is present above the maximum associated with LSW (Fig. 3c, dark blue line). Subsequently, the mixed layer depth decreases to almost zero (Fig. 3b), indicating surface restratification, but a maximum in layer thickness remains. In the profiles (Fig. 3c), this can be seen as lower-density water accumulating above the maximum around $\sigma_\theta = 27.74 \, \mathrm{kg \, m^{-3}}$. The magnitude of the maximum decreases through mixing with surrounding water, but it is still detectable in the profile as the float enters the boundary current.

The evolution of layer thickness and mixed layer depth shown in Fig. 3 is representative of a typical float measuring LSW formation in the interior that later enter the boundary current. Generally, after the initial profile measuring a deep mixed layer, surface restratification and mixing lead to a decrease in maximum layer thickness, but the signature of LSW remains detectable. To ensure that only floats moving with newly formed LSW were used, we discarded data from floats with changes in maximum layer thickness of more than 50% between consecutive profiles. As a result, data from about 12% of all floats measuring convection were omitted from the analysis, but the findings discussed in Sect. 3.1 do not change qualitatively if all float data are used instead. The layer thickness criterion is used only for floats measuring convection in the interior. For convection within the boundary current, LSW input occurs at the time of the initial profile, and we only use the constraint that 2 of the next 3 profiles have to be in the boundary current.

In order to investigate the seasonal timing of LSW input into the boundary current, we define a LSW input rate which is simply given by the number of floats measuring LSW input at a given time, divided by the total number of floats entering the boundary current

$$Input_{LSW}(t) = \frac{n(t)}{N} \tag{1}$$

Where $t$ is the time given as day of year in 5–day bins, $n(t)$ is the number of floats inputted during each time step, and $N$ is the total number of floats measuring LSW input. This is used in Sec. 3.2 to compare the LSW input to the seasonal cycle of oxygen at K9. One possible caveat of this method is that it does not take into account the volume of LSW associated with each instance of input, which would require some information about the horizontal extent of each patch of LSW measured by

the floats. Furthermore, variations in the number of floats present in the region at different times could lead to biases in the resulting estimate of the LSW input rate.

## 3   Results

Time series of oxygen at 600 m nominal depth (Fig. 4) show an overall similar picture at all sites: Starting in February–March, high-frequency oxygen variability intensifies, and mean oxygen concentrations increase by around 10 $\mu mol\,L^{-1}$, remaining elevated during spring. Subsequently, the short-term variability subsides, and beginning in July, oxygen concentrations decrease slowly throughout the year, until the next increase the following winter. The shift in oxygen concentrations appears to occur in concert with changes in temperature and salinity (Fig. 5). T/S changes occur largely along isopycnals, meaning that the effect of T and S variations on density compensate each other. Correlation between T and $O_2$ is high for all of the sites, between -.80 and -.94. Although some of this can be explained by solubility changes, which are dependent on temperature (Garcia and Gordon, 1992), correlations between saturation percentage and temperature are still high, between -0.58 and -0.84. If changes in oxygen were simply due to temperature-driven solubility differences at a constant saturation percentage, correlation of saturation and temperature would be zero. The fact that both saturation percentage and concentration are correlated with temperature suggest that the changes are associated with the presence of different water masses. In fall and early winter, measurements show warmer, saltier and lower oxygen water with lower saturation percentage, while colder, fresher water with higher $O_2$ concentration and saturation is predominantly observed in late winter and spring. The majority of the T/S measurements lie roughly along a mixing line between typical endmembers for LSW and IW (see Fig. 5 for reference). The clustering of measurements between these two endmembers suggests that variability of properties at 53° N, including oxygen, is chiefly controlled by changes in the fraction of each source water mass found in the boundary current.

In the following sections, we will analyze the main features of the time series in more detail, discussing first the initial oxygen increase in February and March and associated higher-frequency variability, using data from the K9 mooring (Sect. 3.1). This is followed by an analysis of a complete seasonal cycle of oxygen concentration at K9, covering convection and the following restratification period in 2016–2017 (Sect. 3.2), and a brief analysis of the differences between the moorings (Sect. 3.3).

### 3.1   High-frequency oxygen variability in February–April

As oxygen concentrations at the mooring sites begin to increase in late February, there is a concurrent increase in the spread of the measurements (Fig. 4). In 2017, during the months of February, March, and April, oxygen concentrations at K9 vary between values typical of the months prior, between 297 $\mu mol\,L^{-1}$ and 301 $\mu mol\,L^{-1}$, and higher values ranging from 305 $\mu mol\,L^{-1}$ to 315 $\mu mol\,L^{-1}$, which subsequently persist until late June. During these months, the spread between the minimum and maximum values measured in a single day can be as much as 15 $\mu mol\,L^{-1}$, and the standard deviation increases from 1.1 $\mu mol\,L^{-1}$ for the 3-month period from 1 November, 2016 to 31 January, 2017 to 4.2 $\mu mol\,L^{-1}$ for 1 February, 2017 to 30 April, 2017.

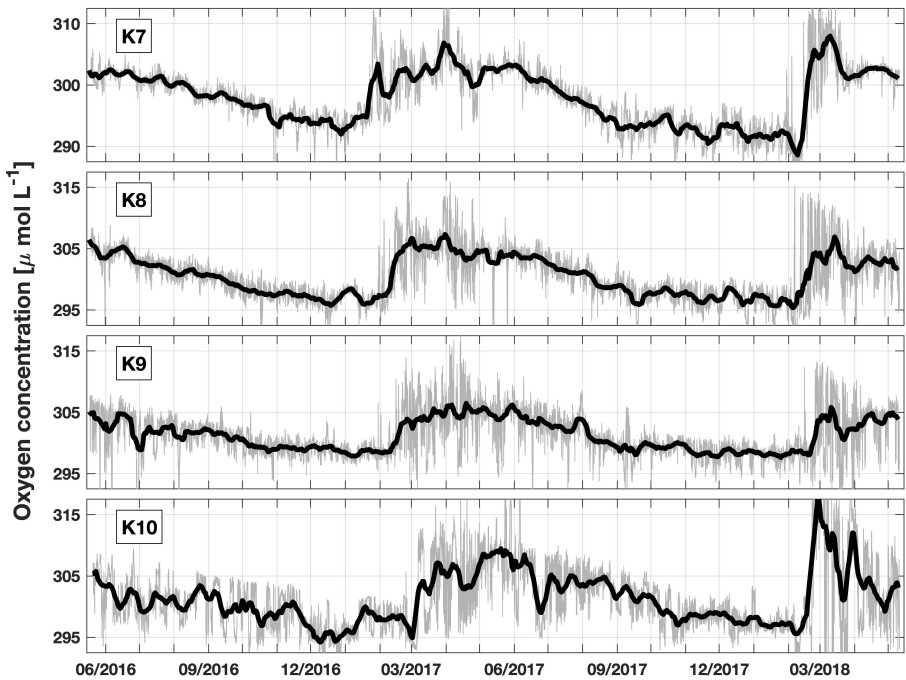

**Figure 4.** Time series of oxygen concentration at 610 m for K7, K8, K9 and K10 (top to bottom). Gray lines show original 15-minute data, and black lines show time series filtered with a 10-day running mean. Note that axis limits are 5 μmol L$^{-1}$ lower in the top panel, but the range is the same for all panels.

This wide spread in the measured oxygen concentrations is also reflected in differing temperature and salinity properties for a 20-day period in February (Fig. 6a). As for the overall dataset, the highest oxygen values occur at the lowest temperatures and salinities. Starting from a cluster of points with higher temperatures and salinities for February 7 to February 12, water with
properties of LSW begins to emerge in the following days, and by February 27, some of the measured T/S values are similar to those found in the center of the basin during convection (see also Fig. 5). During this time, much of the annual range of temperature, salinity, and oxygen properties found in the 2–year record is observed within just 20 days (see Figs. 5, 6a). High-frequency variability in properties at the beginning of the convection season was also observed in temperature measurements further upstream in the boundary current by Cuny et al. (2005). They concluded that LSW was being formed within the
boundary current itself, and attributed the high-frequency variability to spatially varying atmospheric forcing producing LSW with differing properties. However, the fact that there is also a large spread in oxygen in Fig. 6a suggests that some of the water observed during this time has not yet been ventilated. Instead, the variability could be a sign that the earliest convective activity is episodic and patchy, resulting in newly ventilated LSW being transported with the boundary current alongside the lower-oxygen IW found prior to the onset of convection.
Although Cuny et al. (2005) showed that LSW formation can occur in the boundary current, there is no evidence of local convection near 53° N in our data. While no near-surface instrument was deployed on the K9 mooring, data from about 50 m

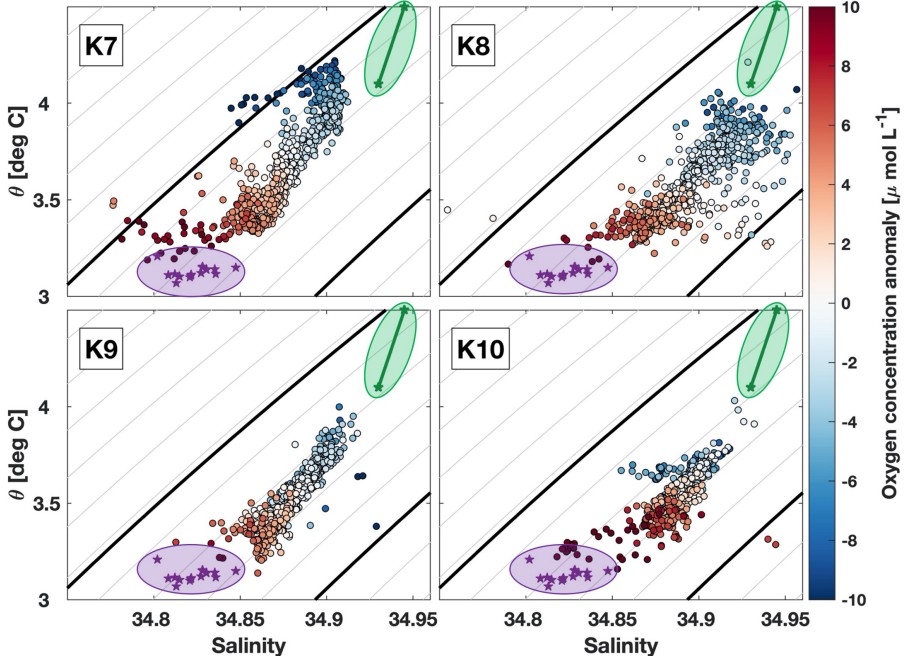

**Figure 5.** Temperature and salinity diagrams at about 600 m nominal depth from all four boundary current moorings, showing daily averages. Colors correspond to oxygen anomalies relative to the mean at each instrument, ranging from -10 μmol L$^{-1}$ (blue) to +10 μmol L$^{-1}$ (red). Purple stars and shading show values from the wintertime mixed layer at the SeaCycler mooring in the interior Labrador Sea (Atamanchuk et al., 2020), and green symbols and shading show typical wintertime conditions of IW near Greenland, taken from Pacini et al. (2020). Potential density ($\sigma_\theta$) contours are drawn with a spacing of 0.025 kg m$^{-3}$, and black solid contours show densities of $\sigma_\theta = 27.68$ kg m$^{-3}$ and $\sigma_\theta = 27.8$ kg m$^{-3}$.

depth at K8 suggest that the boundary current at 53° N remains stably stratified (Fig. 6b). Vertical density gradients generally exceed the $\Delta\sigma_\theta = 0.01 kg m^{-3}$ threshold often used to determine mixed layer depth in the region (Piron et al., 2016), as well as the less stringent threshold of $\Delta\sigma_\theta = 0.03 kg m^{-3}$ used by de Boyer Montégut et al. (2004). This suggests that there is no

active convection near the mooring sites during the study period. Instead, we conclude that the water must be advected to the 53° N array from upstream. The observations are consistent with LSW being formed in the interior of the basin and entering the boundary current along isopycnals (Georgiou et al., 2020). Alternatively, the observed changes could also be explained by convective activity taking place within the boundary current further upstream.

In order to identify the source of the LSW arriving at 53° N in February, we use data from Argo floats to investigate the
timing of convection and input into the boundary current. Fig. 7 shows the fraction of profiles within 50 km of each $0.5° \times 0.5°$ bin that measured mixed layers deeper than 600 m based on the Holte et al. (2017) Argo mixed layer depth database. The most salient feature is a large area of deep mixed layers in the interior of the basin, near the mixed patch often defined as the LSW formation region (Lazier, 1973; Yashayaev, 2007; Atamanchuk et al., 2020). The area extends eastward and indicates a major

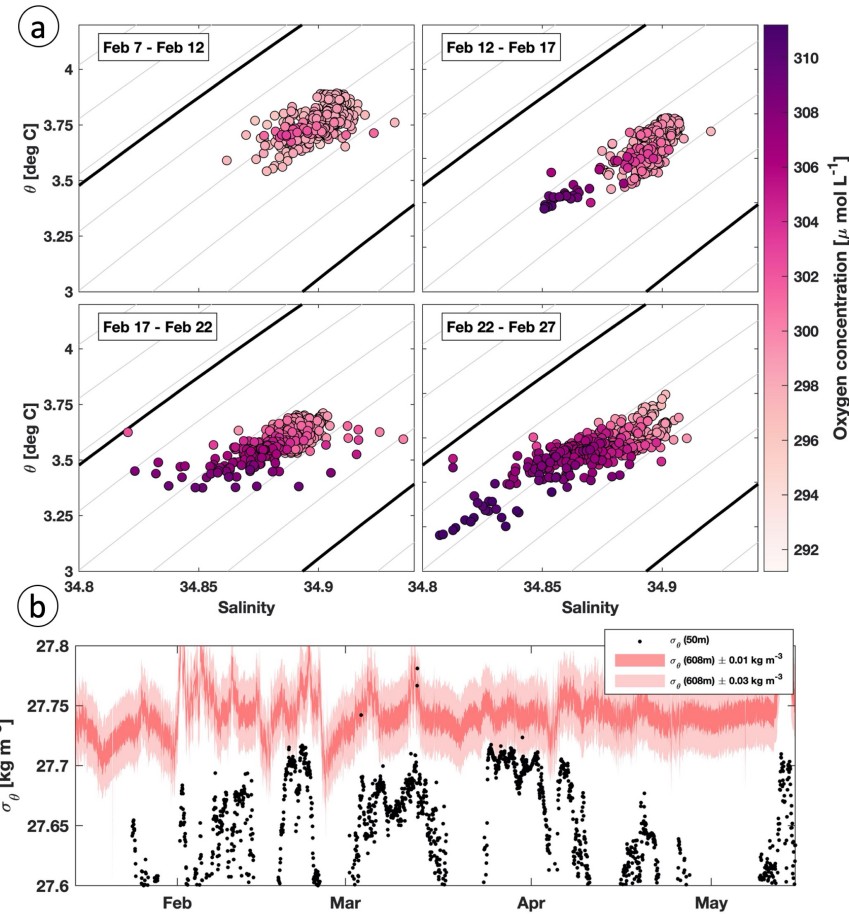

**Figure 6.** a) T/S properties at K9 at 15-minute resolution for four consecutive 5-day periods during the initial oxygen increase in February. Colors correspond to oxygen concentration. Potential density ($\sigma_\theta$) contours are drawn with a spacing of 0.025 $\mathrm{kg\,m^{-3}}$, and solid black contours show densities of $\sigma_\theta = 27.68$ $\mathrm{kg\,m^{-3}}$ and $\sigma_\theta = 27.8$ $\mathrm{kg\,m^{-3}}$. Note changed axes ranges compared to Figure 5.

b) Density measurements at K8 at 50 m and 608 m nominal depth. Shading for 608 m values shows density ranges of $\pm 0.01$ $\mathrm{kg\,m^{-3}}$ and $\pm 0.03$ $\mathrm{kg\,m^{-3}}$ from the measured values.

interior ocean spreading pathway that connects the Labrador Sea and Irminger Sea (Talley and McCartney, 1982; Sy et al., 1997; Fischer et al., 2018; Zunino et al., 2020). Another region of increased occurrence of deep mixed layers is found inshore of the 3000 m isobath near 55° W, close to where Pickart et al. (2002) first reported evidence of convection in the boundary current. No mixed layers deeper than 600 m are found in the boundary current south of 55° N, consistent with our findings from the mooring data that convection did not occur locally at 53° N.

Also shown in the figure are convection locations for all floats from the 2000–2020 period from the Argo data described in Sect. 2.2, using the last profile measuring a mixed layer deeper than 600 m. The points generally line up with the mean picture from the Holte et al. (2017) data, with differences between the two likely occurring due to continued modification during convection. Overall, the majority of the profiles measuring convection in both datasets are in the interior. Most of those floats that measure deep mixed layers inshore of the 3000 m isobath, are located immediately adjacent to the interior patch.

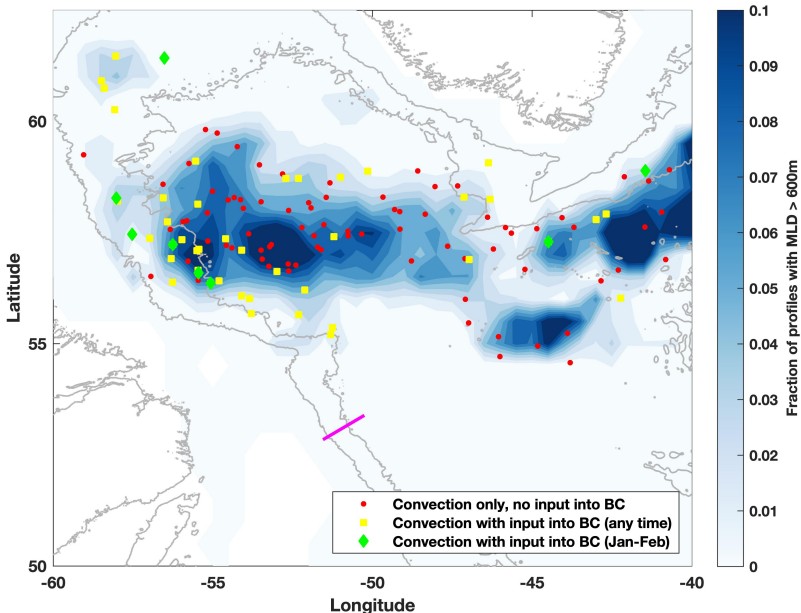

**Figure 7.** Fraction of total profiles from Holte et al. (2017) database with mixed layer depth over 600 m. Grey contours show the 2000 m and 3000 m isobaths, and the magenta line shows the 53° N array. Overlaid are the convection locations for each float measuring convection as defined in Sect. 2.3, differentiating between floats that are not found in the boundary current within the same year (red dots), those that do enter the boundary current (yellow squares), and those that are found in the boundary current in either January or February (green diamonds).

A quantitative analysis of the relative importance of the different formation regions is beyond the scope of this study, as the Lagrangian nature of Argo floats results in undersampling of fast-flowing boundary currents relative to regions with weak currents like the interior Labrador Sea. Nonetheless, our data set reveals some features about the differences between convection in the interior and within the boundary current. The profile locations in Fig. 7 are color coded by whether or not the

float enters the boundary current during the same year, using the definition of LSW input described in Sect. 2.3. About half of the floats measuring convection in the interior remain there until the following convective season, consistent with model studies

showing that newly formed LSW can remain in the interior for several seasons (Straneo et al., 2003; Georgiou et al., 2020). Conversely, most of the floats measuring deep mixed layers inshore of the 3000 m isobath stay within the boundary current afterwards, suggesting that a larger fraction of LSW formed in the boundary current is exported out of the basin compared to LSW formed in the interior. Moreover, out of the more than 100 floats measuring convection to depths exceeding 600 m, only six are subsequently found inshore of the 3000 m isobath in the months of January and February. The convection location

of these floats is highlighted in Fig. 7 by green diamonds. The floats found in the boundary current during this time either measured convection very close to the 3000 m isobath, which allows them to enter the boundary current quickly, or convection within the boundary current region itself. This suggests that the initial arrival of the elevated oxygen signal at the moorings in February is likely to be a result of convection within, or close to, the boundary current.

## 3.2    Seasonal Cycle

To investigate the broader-scale variability, we look at the relation between oxygen and temperature and salinity for the period of December 2016 to December 2017, covering a full annual cycle including convection and restratification. Fig. 8 shows the evolution of the oxygen concentration throughout the year in the form of monthly histograms with a bin size of $1\,\mu mol\,L^{-1}$. The color for each bin shows the mean spiciness ($\pi$), which is a tracer of T and S variations that are orthogonal to density, making it useful for differentiating between water masses with different T/S properties but similar density (Flament, 2002).

Spiciness is used here as a water mass index to quantify the relative contribution of the two endmember water masses, LSW and IW. Fig. 8 uses data from the K9 mooring, and results from all other moorings are discussed in the following section.

Overall, the seasonal cycle is marked by a shift from water with lower oxygen and T/S properties closer to those of IW towards high-oxygen LSW, which is most prevalent in the months of March–July. In January, the bulk of the measurements at K9 show similar properties. The most abundant class of oxygen values is at $298\,\mu mol\,L^{-1}$, and the water mass index shows

higher temperatures and salinities, indicative of IW most likely advected from upstream within the boundary current. There is little spread around this mean value, with almost all measurements falling in the range of 297-301 $\mu mol\,L^{-1}$. In the month of February, measurements showing water with oxygen concentrations around $297\,\mu mol\,L^{-1}$ still make up the majority of values seen in the record, but the number of observations in the central bin decreases. Moreover, a smaller secondary peak emerges at a concentration of $306\,\mu mol\,L^{-1}$, with lower mean temperature and salinity values, and there is a wide spread of oxygen

concentrations measured, ranging from $297\,\mu mol\,L^{-1}$ to $315\,\mu mol\,L^{-1}$. This is consistent with the view that, at this time, the boundary current waters comprise non-ventilated water as well as patches of recently convected and ventilated LSW (see Sect. 3.1).

In the following months, the LSW peak becomes more pronounced and replaces the class of higher temperature and salinity water with low $O_2$ as the most abundant water mass. In April, the most commonly measured $O_2$ value is at its annual maximum

of $306\,\mu mol\,L^{-1}$. By May, the boundary current has become more homogenized, as evidenced by the higher number of observations in the central $O_2$ bin. With the increased input of newly formed LSW, the T/S properties are at the opposite end

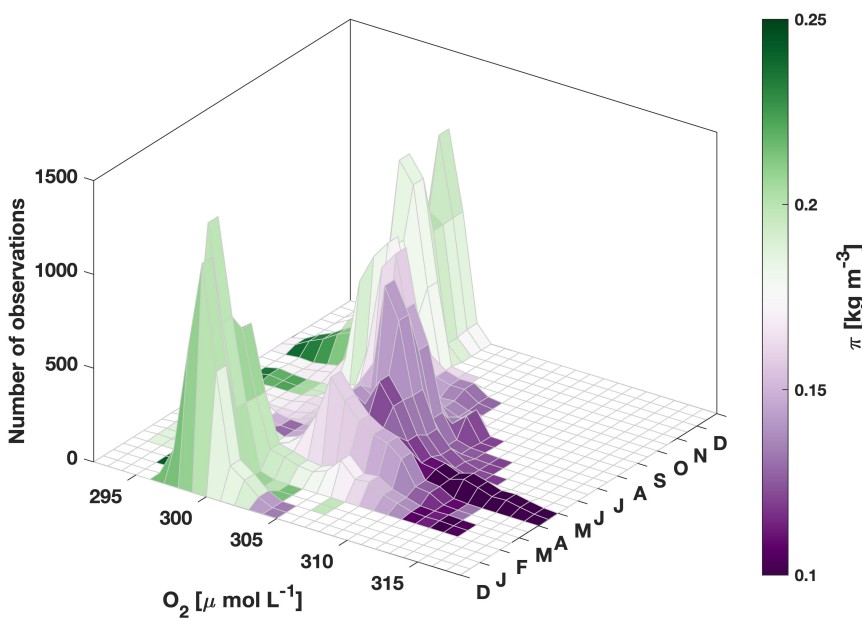

**Figure 8.** Monthly histogram of oxygen from December 2016 to December 2017 at about 600 m depth at K9. Colors show the mean spiciness of each $O_2$ bin, with lower values corresponding to LSW and higher values to IW (see text for definition).

of the water mass index ($\pi$) compared to December–February. After June, oxygen concentrations are beginning to decrease again, with a concurrent increase in T and S, indicative of a larger fraction of IW. Only isolated instances of elevated oxygen concentrations are observed at K9 by August and September, as the water at 600 m depth becomes warmer, saltier, and less

oxygenated. In the autumn months, the signature of LSW has all but disappeared, with water found at the mooring once more resembling the properties of IW, before the cycle begins anew the following winter.

Fig. 9 compares the seasonal oxygen cycle at K9 from Fig. 8 to the seasonal cycle of LSW input into the boundary current, and a partial seasonal cycle at 500 m depth from the SeaCycler mooring in the interior of the basin (see Fig. 1 for location). The input of LSW into the boundary current is an average over all years from the float data used in Sect. 3.1, and is calculated using

equation 1. The gray bars show the LSW input during each 5-day period resulting from convection within the boundary current and LSW entering the boundary current from the interior, respectively, and the black line shows the combined input. Overall LSW input begins to increase in late January, peaks in April, and vanishes in July except for an isolated instance later in the year. Only 47 floats from the data set measure LSW input into the boundary current, which could result in some bias of our estimate relative to the true underlying variability. However, the timing is consistent with related measures such as the amount

of LSW leaving the interior derived from measurements of layer thickness (Yashayaev and Loder, 2016), and rates of boundary-interior exchange inferred from heat content changes (Straneo, 2006), suggesting that the overall variability is captured despite the very limited number of data points. The curve has a similar shape to the seasonal cycle of the most commonly measured $O_2$

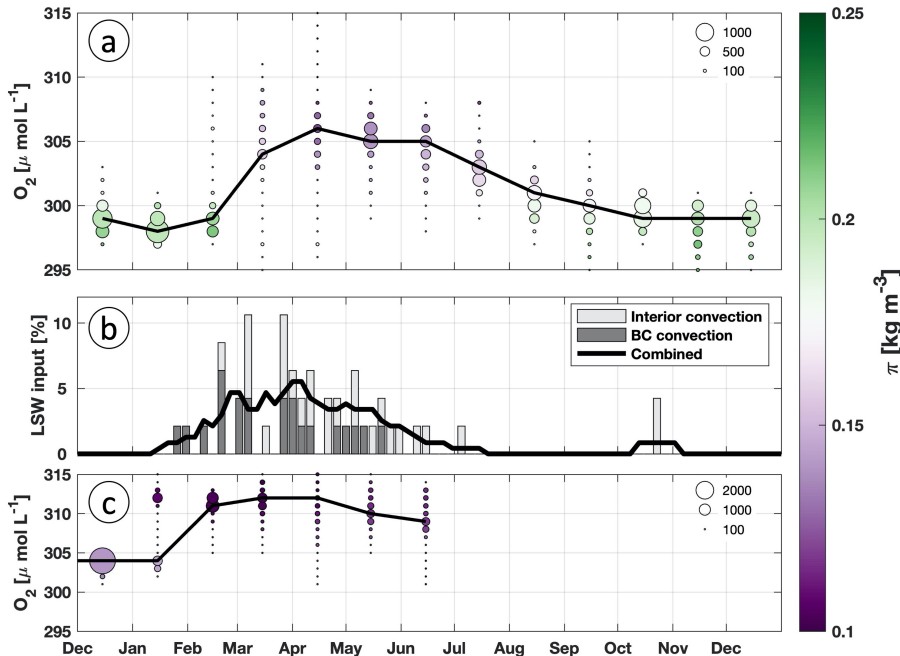

**Figure 9.** a) Monthly histogram of oxygen from December 2016 to December 2017 at 610 m depth at K9, as in Fig. 8. The size of each circle corresponds to the number of observations, the black line shows the bin with the highest number of observations for each month, and colors show the mean spiciness of each $O_2$ bin.

b) Climatological seasonal cycle of LSW input into the boundary current from float data in 5-day bins, as a fraction of the total input of LSW over the year (equation 1). Gray bars show separately the input from boundary and interior convection in each 5-day bin, and the black line shows the combined input smoothed with a 5-point running mean.

c) Monthly histogram of oxygen from December 2016 to July 2017 at 500 m depth for the SeaCycler mooring in the deep convection area in the interior of the basin. The size of each circle corresponds to the number of observations, the black line shows the bin with the highest number of observations for each month, and colors show the mean spiciness of each $O_2$ bin.

concentration at K9 for each month, shown as a black line in the top panel, but the two are shifted relative to one another. The $O_2$ concentration maximum increases in March, reaches its peak in April, and then stays at a similar level until June, before starting to decrease in July and August, lagging the increase in LSW input by about 1–2 months. Typical current speeds in the boundary current at the depth of the sensors used here are on the order of $15\,\mathrm{cm\,s^{-1}}$ (Zantopp et al., 2017), and the distance to the region where the interior convection area is closest to the 3000 m isobath and boundary current convection is most frequent is about 450 km (see Fig. 7). The time scale for LSW to arrive at the 53° N moorings after entering the boundary current would therefore be about 35 days. This is consistent with the time lag of approximately one month between the seasonal cycle of LSW input and the $O_2$ maximum time series at the K9 mooring. The data therefore support the hypothesis that the increase in oxygen concentration at the exit of the basin is largely controlled by the rate of formation and subsequent export of LSW.

LSW input from convection that occurs directly within the boundary current precedes the outflow of LSW from the interior by about a month, supporting the qualitative assessment made in Sec. 3.1 that the initial arrival of newly ventilated LSW at 53° N is likely a result of boundary current convection. Overall, 45% of the input (21/47 floats) is associated with boundary current convection, and 55% (26/47) with interior convection, with both contributing about equally in March and April when overall LSW input is highest. However, due to the caveats of our method discussed in Sec. 2.3, these numbers should not be interpreted as a measure of the LSW volume flux associated with the two sources.

At the SeaCycler mooring, there are two distinct peaks in the oxygen histogram in January, as the mixed layer reaches the sensor depth and ventilates water that had not been in contact with the atmosphere since at least the previous winter. In February, newly ventilated, high-oxygen LSW becomes the most prominent water mass, and $O_2$ remains elevated until the end of the available record in June 2017. The maximum oxygen concentration found in the interior before January is 304 $\mu$mol L$^{-1}$, much lower than the maximum values at 53° N during March–July, confirming that the increased oxygen levels of the outflowing boundary current must be associated with recently ventilated water, rather than LSW formed in previous winters.

If the bulk of the LSW entering the boundary current from the interior originates in the center of the basin near the SeaCycler mooring, the shortest distance to the boundary current would be about 230 km. With a time difference of about a month between the oxygen increase at SeaCycler and the increased input of LSW from the interior into the boundary current, this implies a mean advection speed of 9 cm s$^{-1}$. This is much larger than the climatological advection speed in the interior (see Fig. 1), suggesting that short-lived features such as eddies and current meanders may play a crucial role in the export of newly formed LSW.

## 3.3 Differences at the 53° N array

Although the seasonal oxygen cycle at K9 discussed in the previous section is representative of the general picture along the 53° N array, there are also some horizontal differences. The four moorings that carry oxygen sensors are deployed in different parts of the boundary current system at the exit of the Labrador Sea (Zantopp et al. (2017), their Figure 1). The K9 mooring, discussed in the previous section, is situated within the core of the Deep Western Boundary Current (DWBC), which has only a weak velocity shear between 400 m and 2000 m. K7 sits over the continental shelf beneath the core of the Labrador Current, which is surface intensified, and carries colder, fresher water of Arctic origin. The K8 sensors at 600 m depth are within the DWBC, but closer to the boundary with the LC, and can be within it when the currents are meandering. At all three of these moorings, the flow is southward, with mean velocities over the deployment period between -13 cm s$^{-1}$ and -15 cm s$^{-1}$. In contrast, the K10 mooring is at the boundary between the offshore edge of the DWBC and its northward recirculation. Over the deployment period, there is a weak mean northward flow at about 600 m, with a speed of 3.5 cm s$^{-1}$, but daily mean values are distributed evenly around this mean, ranging from -20 cm s$^{-1}$ to +20 cm s$^{-1}$. The seasonal cycle of oxygen concentrations at all four moorings (Figs. 9a and 10) shows elevated values and water with properties closer to LSW during spring and summer, and water with lower oxygen values resembling IW in autumn and winter.

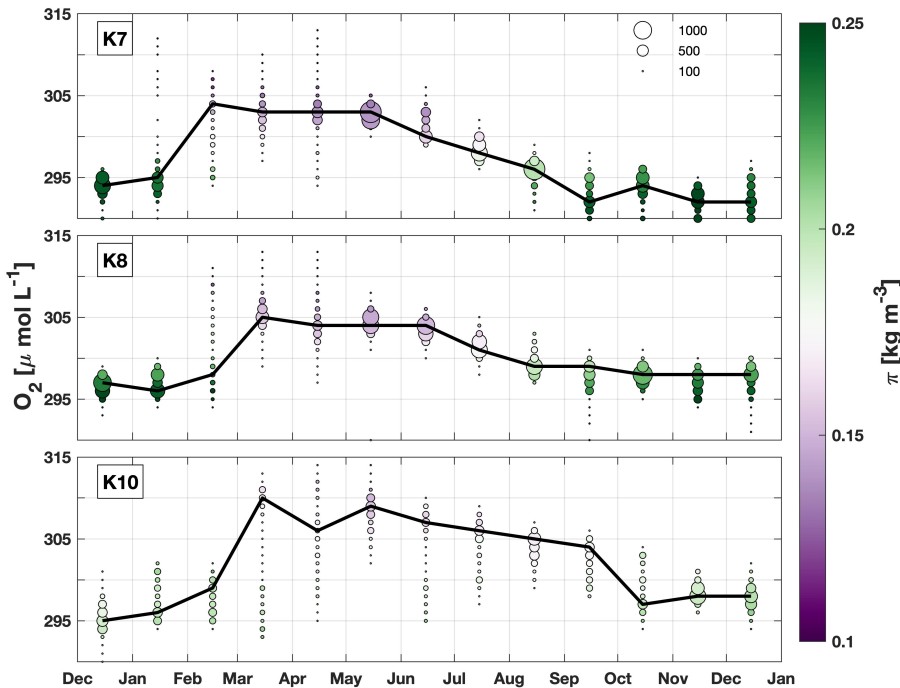

**Figure 10.** Monthly oxygen histograms as in Figure 8a, for the K7, K8, and K10 moorings. Circle size corresponds to number of observations, the black line shows the bin with the highest number of observations for each month, and colors show the mean spiciness ($\pi$) in each bin, highlighting the relative influence of IW and LSW.

At the border of the Labrador Current and DWBC, the oxygen concentrations measured at K8 show a similar seasonal cycle
as those at K9. Properties measured by the water mass index ($\pi$) are also comparable, although the spread in T and S is higher (Fig. 5), suggesting that the DWBC here is less homogenized than at K9.

Over the continental slope at K7, the oxygen concentration is lower by about 5 $\mu$mol L$^{-1}$ throughout the year, and the earliest arrival of elevated oxygen occurs in January compared to February at K9. The water is generally warmer and fresher than at the other moorings, which may be the reason for the lower oxygen values, and also results in a slightly lower mean density.
There also seems to be a stronger seasonal variation of density compared to the other moorings. Between the warmer, saltier, oxygen-poor water in winter and the high-oxygen, low T/S water in spring, density increases by more than 0.05 kg m$^{-3}$ (Fig. 5). This is similar to variability observed upstream in the boundary current near 55$°$ N by Cuny et al. (2005), who showed that this is a result of isopycnals steepening in winter over the continental slope. This weakened stratification may allow for the formation of uLSW (Pickart et al., 1997). At K7, the highest oxygen concentrations occur at lower densities and salinities, seen
as a "tail" in the bottom left corner of the T/S plot. Moreover, the first high-oxygen measurements occur almost three weeks earlier than at K9, which may suggest that these are a product of formation of uLSW within the Labrador Current, and that this formation takes place closer to the mooring section than the production of classical LSW found at the other moorings. Since

Argo floats generally stay offshore of the 2000 m isobath, formation of uLSW within the Labrador Current would be difficult to detect with the methods used in Sect. 3.1.

At the offshore edge of the southward boundary current, at mooring K10, the oxygen concentrations are more spread out, and elevated values are not found until the beginning of March. The spread of oxygen concentrations measured during any given month is much larger than for the other moorings, and the T/S signature of the two different water masses is less pronounced. This is likely a result of current meanders, with the mooring measuring the DWBC some of the time and its recirculation at other times, resulting in a wider range of ventilation histories. High oxygen concentrations are found during times of both

southward and northward flow, suggesting that LSW is carried southward with the offshore edge of the DWBC at K10, but some of it may also enter the northward recirculation back towards the Labrador Sea.

## 4   Discussion

### 4.1   LSW formation and export pathways

The analysis in Sect. 3.1 shows that deep mixed layers associated with convection are more commonly found in the interior.

In Fig. 7, the area where more than 5% of profiles measure mixed layers deeper than 600 m is about 15 times larger in the interior than inshore of the 3000 m isobath. However, this does not necessarily translate to 15 times larger oxygen supply, as water parcels have a longer residence time in the interior, whereas newly formed LSW in the boundary current is rapidly exported (MacGilchrist et al., 2020, 2021). This is also evident in our data analysis, where a significant number of Argo floats measuring deep mixed layers in the interior do not enter the boundary current during the same season, and therefore do not

directly contribute to the oxygen export by the outflowing boundary current. Moreover, LSW formation within the boundary current can also occur as slantwise convection (Cuny et al., 2005), or in parts of the boundary current that are shallower than 2000 m, neither of which would be detectable with Argo floats. As discussed in Sect. 3.3, some evidence of convection over the continental slope upstream can be seen in data from the K7 mooring, at a water depth of 1400 m. However, neither the mooring data, which do not measure local convection, nor the Argo data, biased towards sampling regions with mild currents at

depth, provide sufficient information to quantitatively determine the portion of the LSW exported at 53° N that is formed in the boundary current. The importance of boundary versus interior convection for the outflowing LSW remains an open question, with one previous estimate from float data concluding that both could be important for the properties of the outflowing boundary current (Palter et al., 2008).

       Another crucial aspect concerning the export of LSW from the formation region is how the water formed in the interior

enters the boundary current. Palter et al. (2008) showed that both eddies and a mean cross-isobath flow can contribute to LSW export, with floats entering the boundary current on all sides of the basin, which is also the case for the float data used here (not shown). The time scale for a water parcel to round the basin from Cape Farewell to the Labrador side is about 147 days (Cuny et al., 2002). This implies that LSW entering the boundary current, or formed within it, on the northern side of the basin during the height of convection in March would not reach 53° N until August, when the influence of LSW at K9 is already decreasing.

Thus, the outflow of LSW during summertime observed at 53° N is likely fed mostly by LSW entering the boundary current

on the western or southwestern side, allowing for faster export out of the basin. The timing of the seasonal cycles of oxygen concentration in the interior, LSW input from the interior into the boundary current, and oxygen concentration at K9 (Fig. 9) implies that eddies may play an important role in exchanging water between the boundary current and interior, consistent with previous studies (Eden and Böning, 2002; Straneo, 2006; de Jong et al., 2014).

In a recent study, Georgiou et al. (2020) used model data to analyze the source of water parcels reaching the 53° N section, and found that 41% did not leave the boundary current after passing Cape Farewell at the southern tip of Greenland, while a further 44% are modified in the interior of the basin before re-entering the boundary current, and the remainder follow several less common pathways. These numbers are broadly consistent with the results from Fig. 8, with about equal amounts of LSW and IW found at the moorings over an annual cycle. Our study suggests that there may be a seasonal dependence to the split observed by Georgiou et al. (2020), with the LSW export (interior route) occurring chiefly in the months after convection, and IW (water from Cape Farewell) being more ubiquitous during autumn and winter. The input of newly ventilated LSW into the boundary current in the months following convection observed here is consistent with the more direct southward export of lighter LSW found in their study, as well as results from an earlier modeling study (Brandt et al., 2007). Georgiou et al. (2020) suggested that pathways are different for the densest LSW formed in the interior, with water entering the boundary current on the northern side of the basin, thus taking longer to reach 53° N. This density dependence may explain why most of the LSW found in the mooring record seems to follow the faster southward export route, as the lower part of the LSW layer is not sampled in the current sensor setup. Another model study by Handmann (2019) highlighted different routes that exist for LSW export, including several pathways for LSW entering the Irminger Sea. Their study also proposes that some of the water exported through the 53° N section subsequently leaves the DWBC at Orphan Knoll near 50° N and recirculates back towards the north, reaching both the Labrador and Irminger Seas. In our data from the mooring furthest offshore, K10, there is evidence of high-oxygen water with northward velocities (Sect. 3.3), suggesting that newly formed LSW may also be entrained into this northward recirculation.

These studies show that many more LSW export pathways exist in addition to the direct southward route that seems to be responsible for the pronounced seasonal signal at the 53° N moorings. Although the large summertime increase in oxygen (Fig. 8) is associated with LSW formed in the same year, the oxygen concentrations can also be impacted by export of LSW along these alternative routes, as well as further water mass transformation along the pathway. For example, some of the LSW formed in the interior one year may be advected into the Irminger Sea, enter the boundary current on the eastern side of Greenland, and stay within it until it reaches 53° N several years later. Deep convection also takes place in the Irminger Sea (de Jong et al., 2018), leading to a somewhat weaker, but still significant uptake of oxygen (Maze et al., 2012; Palevsky and Nicholson, 2018). The water mass formed by convection in the Irminger Sea, known there as Irminger Sea Intermediate Water, or ISIW (Le Bras et al., 2020), is mixed into the boundary current in the Irminger basin before it enters the Labrador Sea, affecting the properties of what we have referred to here as Irminger Water (IW). Therefore, both the export of LSW into the Irminger Sea and convection occurring there will have affected the properties of the "low-oxygen" IW seen in Fig. 8–10, and the annual minimum concentration of 298 $\mu\mathrm{mol\,L^{-1}}$ at K9 is likely already elevated compared to what it would have been without ventilation in the Labrador and Irminger Seas.

## 4.2 Global impact of Labrador Sea ventilation

Due to the global importance of LSW, it has long been assumed that changes in its formation would play a crucial role in setting the variability of the Atlantic Meridional Overturning Circulation (Koltermann et al., 1999; Yashayaev, 2007). However, results from the Overturning in the subpolar North Atlantic program (OSNAP) have called into question this long-held belief, showing that the bulk of the water mass transformation across isopycnals occurs east of Greenland in the Irminger Sea and Iceland Basin. This water mass transformation in the eastern section appears to be the main source of overturning variability (Lozier et al., 2019), although recent modelling results suggest that LSW formation may still be important for variability on multidecadal time scales that are not yet resolved by the OSNAP time series (Yeager et al., 2021).

In a recent follow-up study to the OSNAP results, Zou et al. (2020) showed that the disconnect between overturning and LSW formation is in large part due to density-compensated changes in temperature and salinity. Although over 12 Sv of water mass transformation occurs in the Labrador Sea from warm, salty waters to colder and fresher ones, the net transformation into the LSW density range is only about 4 Sv. Our results are consistent with these findings, showing that while the water mass properties of the outflow are altered substantially by the input of newly formed LSW, its density remains largely unchanged. This suggests that LSW from the interior enters the boundary current primarily along isopycnals, consistent with recent results from models (Brüggemann and Katsman, 2019; Georgiou et al., 2020), with some along-isopycnal mixing occurring before it reaches 53° N. Our time series also show for the first time direct evidence that significant changes occur in the oxygen content of the outflow, driven by the rapid export of newly ventilated LSW. While we only use data from a fixed depth, oxygen concentrations in the summer are elevated in much of the LSW layer above 1200 m (Fig. 2). Assuming that the oxygen input is comparable throughout this layer, we can calculate a first estimate of the oxygen transported out of the basin due to the input of newly formed LSW: Using a mean transport for the LSW layer at 53° N of $14.5 \pm 3.8$ Sv (Zantopp et al., 2017), the oxygen increase of 6 $\mu mol\,L^{-1}$ at K9 during March–August 2017 relative to the January baseline of 298 $\mu mol\,L^{-1}$ (Fig. 8) corresponds to an oxygen export of $1.37 \pm 0.37$ Tmol (1 Tmol $= 10^{12}$ mol) for these six months. Over the whole year, the mean oxygen increase is 3.5 $\mu mol\,L^{-1}$, corresponding to an export of $1.60 \pm 0.42$ Tmol relative to the baseline value. With an integrated wintertime uptake across the air-sea interface due to gas exchange of 21.2 $mol\,m^{-2}$ (Atamanchuk et al., 2020) and a convection area of approximately 150,000 $km^2$, the total uptake in the interior is 3.18 Tmol during the same year. Thus, about 43% of the oxygen taken up during convection is exported out of the basin via this fast export route of LSW along the DWBC during summer, and another 7% through slower export routes during the rest of the year. The good agreement with the air-sea flux estimates from Atamanchuk et al. (2020) provides independent support for their notion that bubble injection plays an important role in oxygen uptake during deep convection. As discussed in Sect. 4.1, LSW is also exported along different pathways than we considered here, and some remains in the basin until the next year, which may account for the remaining 50% of the oxygen uptake. Conversely, the recirculation of the DWBC towards the formation region after it reaches Orphan Knoll (Fischer and Schott, 2002) may diminish the net oxygen export, although the recirculation strength is typically only about 10% of the DWBC transport at 53° N (Zantopp et al., 2017).

The deep North Atlantic is one of the few ocean regions that has not experienced significant deoxygenation in recent decades
(Schmidtko et al., 2017), with oxygen increasing slightly by about 1.7 $\mathrm{Tmol\,year^{-1}}$ over their multi-decadal study period,
compared to a globally integrated oxygen loss of 70 $\mathrm{Tmol\,year^{-1}}$ below 1200 m depth. The continued formation and export
of LSW and NADW may have contributed to staving off deoxygenation in this basin, and the 1.60 $\mathrm{Tmol\,year^{-1}}$ of export
calculated for 53° N could be sufficient to supply much of the oxygen consumed in the uNADW layer in the North Atlantic.
Using a global mean apparent oxygen utilization rate of 0.1 $\mathrm{\mu mol\,kg^{-1}\,year^{-1}}$ at 1500 m depth (Karstensen et al., 2008), an
area of 26,159,000 $\mathrm{km^2}$ for the deep Atlantic Ocean between the equator and 50° N, and a mean layer thickness of 800 m
for the isopycnal range $27.68\ \mathrm{kg\,m^{-3}} \leq \sigma_\theta \leq 27.8\ \mathrm{kg\,m^{-3}}$, we estimate the annual oxygen consumption for this volume of
water to be 2.2 $\mathrm{Tmol\,year^{-1}}$. If the depth-varying profile from Karstensen et al. (2008) is used instead, with the depth range
occupied by uNADW at each point estimated from density data, the estimate increases to 3.8 $\mathrm{Tmol\,year^{-1}}$. Using these two
estimates as a range of possible values for the annual oxygen demand in this volume, the rapid southward export of LSW
discussed here supplies 42–73% of it, with the rest likely provided by slower export of LSW through other routes, as well as
uptake in other deep water formation regions such as the Irminger Sea (Palevsky and Nicholson, 2018), Iceland basin (Maze
et al., 2012), and the Gulf of Lion in the Mediterranean Sea (Ulses et al., 2021). Thus, despite playing a less significant role
than previously thought in setting the strength of the overturning (Lozier et al., 2019), convection in the Labrador Sea does
appear to crucially contribute to the supply of oxygen from the subpolar North Atlantic to the deep subtropical and tropical
Atlantic Ocean. This highlights the important distinction between the overturning circulation and the ventilation of deep waters,
as discussed by Naveira Garabato et al. (2017) and MacGilchrist et al. (2020, 2021). Although the density transformation is
weaker in the Labrador Sea, there is a more direct connection between the atmosphere and deep ocean during the formation
of LSW compared to the water mass formation and transformation processes in the eastern subpolar gyre, allowing for more
intense oxygen uptake.

## 4.3 Outlook

The focus of the present study was to analyze the seasonal cycle in 2017, the one full calendar year that was resolved by
our two-year data set. Although only data for the first half of the next year is shown in Fig. 4, some interannual changes in
the arrival of LSW in 2018 are already apparent, such as a larger oxygen increase at K7, and a large increase followed by
stronger short-term variability at K10. The formation and properties of LSW are well established to be highly variable in time
(Lazier, 1973; Yashayaev, 2007), and our data were collected during a period which featured some of the deepest reaching
convection since the 1990s (Yashayaev et al., 2020). Although current meter data from the 53° N array show no clear relation
between volume transport and the strength of deep convection (Dengler et al., 2006; Fischer et al., 2010), it might intuitively be
expected that weaker LSW formation would lead to less LSW input into the boundary current, and consequently lower oxygen
concentrations. Accordingly, the oxygen export during a year like 2010 with some of the shallowest convection in the interior
(Yashayaev and Loder, 2016) may be weaker than the values reported here for 2017. The extent to which the oxygen export
estimates presented here are subject to interannual variability will be addressed in a future study, using data from continued
deployments of the oxygen sensors at 53° N that will extend the time series until 2022.

A key topic of research regarding LSW formation is its susceptibility to future climate change. Model studies have suggested that increased freshwater input resulting from the melting of Greenland's ice sheets may weaken or even shut down convection in the Labrador Sea (Manabe and Stouffer, 1997; Rahmstorf, 1999; Clark et al., 2002). While the re-emergence of LSW formation to near record depths in 2014–2018 shows that the effect of ice melt has not yet critically affected convection (Yashayaev and Loder, 2016; Yashayaev et al., 2020), recent climate projections and observations continue to point towards a possible shutdown in the future (Böning et al., 2016; Oltmanns et al., 2018). If deep convection in the Labrador Sea were to cease, the "trap door" of oxygen uptake to the deep ocean would be shut (Atamanchuk et al., 2020), cutting off a large part of the oxygen supply discussed above. Moreover, since ventilation of LSW can occur without significant density-space overturning, large changes in oxygen supply are possible even if the overturning circulation remains stable.

In a recent model study on future deep ocean deoxygenation, Oschlies (2021) posited that their work *"calls for more research efforts to explore the baseline of these systems before the unavoidable change will hit"*. The present study represents a first step towards quantifying the impact of LSW formation and export on the supply of oxygen to the open Atlantic Ocean. However, more measurements will be needed to better understand the complex interplay between different deep water formation regions, and the resulting supply of oxygen associated with ventilation in the subpolar North Atlantic as a whole. To that end, oxygen sensors were added to several other moorings in the OSNAP array in 2020, and a total of 70 $O_2$ sensors are now deployed along the path of the boundary current in the Irminger Sea, the Labrador Sea near the southern tip of Greenland, and on the $53°$ N array. With the resulting data, it will become possible to gain a much more complete understanding of the processes controlling oxygen supply from the subpolar North Atlantic than has been presented here.

## 5  Conclusions

The results put forth in this study show the effect of Labrador Sea Water formation and export on the oxygen concentration of the outflowing deep western boundary current. During autumn and winter, properties measured at the $53°$ N array at about 600 m nominal depth are similar to those of Irminger Water, which enters the basin from the Irminger Sea. Following the onset of deep convection, LSW first arrives at the offshore moorings in the second half of February, with a wide range of temperature, salinity, and oxygen properties reflecting a sporadic input of heterogeneous LSW formed in or near the boundary current. As convection becomes more vigorous and sustained, and more newly formed LSW enters the boundary current from the interior, it becomes the dominant water mass found at $53°$ N in April–May. As a result, the most commonly measured oxygen concentration for these months is 7-8 $\mu mol\,L^{-1}$ higher than in December and January. By September, the signature of LSW has largely disappeared, and properties once again resemble those of IW, with accordingly lower oxygen concentrations. This variability in properties seems to be controlled mostly by a direct southward export of newly ventilated LSW with the outflowing boundary current, which occurs chiefly in the 6 months following convection. However, additional measurements throughout the LSW density range would be needed to evaluate whether the seasonal cycle is the same for the entire layer, or just for the part of it measured with the current sensor setup.

The increase in oxygen content due to the input of LSW into the boundary current was estimated to lead to an added oxygen export out of the basin of $1.60\pm 0.42\,\mathrm{Tmol\,year^{-1}}$, which is about half of the oxygen taken up during deep convection in the interior. The annual oxygen supply from this fast, direct export of LSW to the Atlantic Ocean is estimated to be equivalent to 42–73% of the oxygen demand by organisms in the upper NADW layer of the North Atlantic Ocean, highlighting the crucial role that the Labrador Sea plays in the supply of oxygen to the deep ocean. When taking into account the complex recirculation

and export pathways of LSW through the subpolar gyre, as well as uptake in other deep water formation regions, the total added oxygen export associated with North Atlantic Deep Water formation is likely even higher.

     Although the important role of LSW formation for oxygen uptake has long been established, this is the first study that shows a continuously measured time series of the oxygen content of the boundary current over a full year. Likewise, while variations in LSW input into the boundary current over an annual cycle had been reported before, we present the first evidence

that these seasonal variations control the properties of the boundary current at the exit of the basin at 600 m depth. Together with the estimates of oxygen export presented above, these results address the question of how quickly and efficiently newly ventilated LSW is exported, which is particularly crucial in the context of estimating storage rates of gases such as oxygen and anthropogenic carbon. However, our study is only a first step towards answering these questions, and more research will be needed to fully quantify the importance of the subpolar North Atlantic for supplying oxygen to the global oceans via the export

of NADW, and to determine how susceptible this process could be to climate change.

*Data availability.* The mooring and hydrographic data used in this study will be made freely available on pangaea.de. Argo float data were collected and made freely available by the International Argo Program and the national programs that contribute to it. (https://argo.ucsd.edu, https://www.ocean-ops.org).

*Author contributions.* The manuscript was prepared by JKo, with contributions from all coauthors. DA was responsible for designing the

experiment and calibrating the mooring data, and data analysis was carried out by JKo. DW and JKa acquired funding for the project.

*Competing interests.* The authors declare that they have no conflict of interest.

*Acknowledgements.* JKo was funded by the Ocean Frontier Institute through the International Postdoctoral Fellowship. This work was funded in part by the Canada Excellence Research Chair in Ocean Science and Technology. This research was undertaken thanks in part to funding from the Canada First Research Excellence Fund, through the Ocean Frontier Institute. The 53° N array is supported by EU H2020

Blue Action (grant #727852), EuroSea (grant #862626), and the German Ministry of Education and Research (RACE Program). The 53° N array is part of the OceanSITES Eulerian time series network. Argo float data were collected and made freely available by the International

Argo Program and the national programs that contribute to it. (https://argo.ucsd.edu, https://www.ocean-ops.org). The Argo Program is part of the Global Ocean Observing System. The authors would like to thank Marilena Oltmanns for help with processing the mooring data.

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
