# Peer review of "Oxygen export to the deep ocean following Labrador Sea Water formation"

_Biogeosciences, 2021_

## Referee Comment (RC2)

Review on "Oxygen export to the deep ocean following Labrador Sea Water formation" by Koelling et al.

**General comments**

Using moored data at the exit of the Labrador Sea (~53°N), Koelling et al. reported the seasonal variability of oxygen concentration in the outflowing boundary current. The origin of the oxygen signal in the boundary current was further discussed by comparing property fields between the boundary current and basin interior, and by tracking spreading pathways of newly-convected Labrador Sea Water (LSW) with Argo floats. Finally, the importance of Labrador Sea convection was emphasized by estimating the oxygen supply from the Labrador Sea to the deep North Atlantic.

I find the manuscript well-written and the focus on oxygen export is of interest to many people in physical oceanography and biogeosciences. Even so, I still have one major suggestion. The Labrador Sea has been extensively studied, both in terms of the boundary current variability and the spreading pathways of LSW. Some of the findings in the current study, such as the seasonality and LSW pathways, are similar to previous work. Besides the oxygen export estimates, what new/different information is revealed by the current study? Specifying the novel findings in a Conclusion paragraph will greatly enhance the importance of this work.

Below I list specific comments.

**Specific comments**

[1]. "The export of oxygen from subpolar gyre is ~71% of the oxygen consumed annually in the upper NADW layer in the Atlantic Ocean between EQ and 50N". This number was only roughly estimated in the Discussion and should not be listed as a key point in the Abstract.

[2]. Lines 62-64: I believe the importance of LSW in supplying oxygen to global ocean is well known, as stated at the beginning of the Introduction. I suggest the authors to refine their conclusions and implications by specifying the novel findings of the study.

[3]. Figure 2 caption: Please check "$\sigma_\theta$".

[4]. Lines 88-89: Please include 27.74 isopycnal in Figure 2.

[5]. Line 94: It may not be appropriate to call the saltier waters in the Labrador Current as "remnant of Irminger Water" since the waters are significantly mixed. "Remnant" sounds like that a part of Irminger Water is not mixed but retains in the boundary current. Maybe you can simply say it is a mixture of Irminger Water and convective water.

[6]. Line 116: You are defining the export from interior Labrador to the boundary current, instead of the export out of the Labrador Sea (stated in Line 110). Floats that once exported to the boundary current (and stayed for at least 2 cycles) could possibly re-enter the basin interior. A more rigorous selection would be to choose floats that entered the boundary current and stayed there until they exited the Labrador Sea (53N). That would give a smaller number of usable floats but appear to be more appropriate.

[7]. Figure 3 caption: Suggest using "convective interior" or "interior" instead of "convection".

[8]. Figure 4: Please make y-axis consistent between panels. The increase of oxygen in March 2018 is quite sharp compared to March 2017. Could you please include a few comments on that?

[9]. Figure 5: Why are some high oxygen profiles (~10 $\mu mol L^{-1}$) associated with relatively high temperature (~3.5C) and salinity (~34.9) at K10? Does that indicate a different source/route of the oxygen input?

[10]. Lines 175-176: If convection occurs in the boundary upstream of 53N, wouldn't the density anomalies propagate downstream to the array? Do you see that density signal at 53N?

[11]. Line 211: I do not think everyone is familiar with spiciness. I suggest including a paragraph describing how the variable is calculated.

[12]. Lines 238-240 & Figure 9b: If you separate the exported LSW from convection in the boundary current, will the temporal variability of LSW input be different?

[13]. Lines 246-248: I do not think I fully understand this statement. Could you elaborate?

[14]. Line 252: The 12-month time series of LSW input is heavily smoothed (5 months). The 1-month lag could be entirely artificial.

[15]. Line 266: What is "the boundary with of the LC"?

Line 273: What is "the border of the LC and DWBC"? I suggest showing the velocity structure superimposed with the mooring locations for better illustration.

[16]. Lines 290-291: This contradicts Figure 2, which shows smaller oxygen concentrations at K10 compared to the other moorings.

[17]. Line 349: I do not think ISIW, the locally convected water in the Irminger Sea, is the same as Irminger Water. The IW is the saltier and warmer Atlantic-originated water that flows in the boundary current. What enters the Labrador Sea is likely a mixture of ISIW and IW.

[18]. Lines 377-378: I think this is a very intriguing estimate. If recirculation at K10 is considered, how will the (net) export percentage change?

[19]. Line 434: Again, recirculation is not considered and the percentage (71%) could be an overestimation.

---

## Author Comment (AC3)

**Labrador Sea oxygen export and North Atlantic oxygen demand estimates**

Reviewers 1 and 3 both pointed out that the intrinsic uncertainty of the back of the envelope calculation of oxygen demand in section 4.2 was not sufficiently addressed.

To add an uncertainty to the oxygen export estimate, we will change our calculation as follows: Rather than using the current speed, section width, and layer thickness for the calculation (lines 372-373), we will use the published value of mean LSW layer transport from Zantopp et al. (2017), $14.5 \pm 3.8$ Sv, multiplied by the LSW concentration change and integrated over time. Note that this corresponds to only a small change in transport, as the currently used values are equivalent to 14.25 Sv, and the corresponding annual O2 export estimate would slightly increase from 1.57 Tmol/year to 1.60 Tmol/year. Using the Zantopp et al value allows us to use their uncertainty value of 3.8 Sv to add an uncertainty to the oxygen export estimate, which would correspond to 0.42 Tmol/year, or 0.21 Tmol for the summertime increase

Another comment was that there may be issues with using a single deep ocean value for aOUR and a mean layer thickness of uNADW to estimate the oxygen demand (lines 386-389). To address this, we repeated the same calculation using the profile of aOUR as a function of depth obtained from the Karstensen et al. (2008) reference (their equation 3) instead of a constant value. We calculated the vertical extent of the uNADW layer at each grid point, and vertically integrated the aOUR calculated from the equation. The integral of this aOUR estimate over the Atlantic Ocean from 0-50N is 3.79 Tmol/year, which would bring the contribution of LSW down to 42% of the annual oxygen demand.

We will include this estimate in the revised manuscript along with the original calculation, to show the range of possible values based on the assumptions.

**LSW export calculation**

Reviewer 2, 4, and 5 all had concerns or questions about the methodology and implications of calculating export from Argo floats, including

- Definition of export: Does the method of requiring 2 of the next 3 profiles to be in the boundary current capture export, or could floats later re-enter the interior, leading to an overestimate of export? Would a better criterion be to only select profiles that subsequently exit the Labrador Sea in the boundary current through the 53N section?
- How is the "input" of LSW from Figure 9b defined?
- Does the smoothing of the time series lead to artificial peaks that are not based on the actual data?
- Is there a difference in timing between boundary current and interior convection?

These questions are addressed in the discussion and figures below:

[Figure]

*New version of Figure 9, including a curve showing a stricter export criterion*

The figure above shows a revised version of Figure 9, adding to the central panel a second line of export calculated with the same criterion (2/3 following profiles in the boundary current) but also requiring the float to later be exported south of 53N, compared to the original curve. Adding this additional constraint further reduces the number of floats used in the calculation from 47 to 24 but does not significantly change the resulting LSW input estimate.

We propose leaving the analysis as in the original manuscript, but noting in the methods section that results would be similar if using a stricter criterion, and/or showing the line in the figure as shown above.

[Figure]

*Figure showing the curve from Figure 9b, along with original histogram in 5-day bins (gray bars), and individual curves for boundary current and interior convection.*

The input in Figure 9b is defined simply as the fraction of total exported floats that enter the boundary current during a certain time. 47 floats are considered to be exported, so a 5-day period with 5 exported floats would correspond to 10.6% LSW input, and a 5-day period with one exported float would be 2.1%. The resulting time series in 5-day bins is then smoothed with a 5-point (20-day) running mean to arrive at the curve shown in the figure. The figure above also shows the original data in 5-day bins, which we could include in the revised manuscript to better represent the timing of the input.

The red and blue lines show the difference in timing between the input of LSW originating in the boundary current and interior. The two curves are similar in shape, but shifted by about 1 month relative to each other, suggesting again that more of the early export is due to boundary current convection, while the interior contributes more strongly towards the end of the convection period. Peak total export occurs during March and April, when both are near their maximum.

While the exact numbers are likely uncertain due to the nature of the dataset and limited number of floats, this further underlines the distinction between boundary current and interior convection, and is a result that could be of interest to the community. We therefore propose to include the additional curves shown above in the revised manuscript to figure 9b.

---

## Author Response (AR1)

Dear reviewers, Dear editor,

Thank you again for your comments on our manuscript titled "Oxygen export to the deep ocean following Labrador Sea Water formation" submitted to Biogeosciences. We appreciate the constructive criticism you provided, and feel that the resulting changes to the manuscript helped further strengthen its scientific merit. We hope that all suggestions were adequately addressed, and look forward to your next response.

Below you will find our responses to the reviews. Reviewer comments are shown in italic, and our responses from the initial round of author comments are shown in normal text. For more substantial comments, the actual changes made in the revised manuscript are highlighted as green text, and line numbers for the marked-up manuscript are provided.

Kind regards,

Jannes Koelling

**Reviewer 1**

**Specific Comments**

*1) I have some technical comments about some of the figures which I will address later, but I do have a specific comment about Figure 8. Figure 8 and Figure 9a show exactly the same thing. I find unnecessary to have two figures which shows the same, considering that the paper is actually long. I am not asking to short the paper. I do not have a problem with that but reducing the number of figures is already an improvement. Everything that is described in session 3.2 can be easily and actually better follow by looking at Figure 9a. Figure 8 might look fancy but 3D figures are difficult to look on a 2D plane, there is always some part which is hidden. But I could see all the points you described much easier in Fig.9a. I then suggest to remove this figure and refer directly to Figure 9a for the seasonal cycle in session 3.2*

We understand the concern about showing duplicate information in Figure 8 and 9a. However, we believe that having figure 8 is valuable because the arrival of LSW at 53N can be seen more intuitively. We were also hoping to use Figure 8 as the highlight figure shown on the article page on the website, which may not be possible if it is not shown in the paper itself.

*2) Doesn't make more sense to compare the calculated export of 1.57 Tmol instead of with the global oxygen loss of 70 Tmol year⁻¹ with the value calculated only for the deep Atlantic. I think it is an oxygen gain of about 1.7 Tmol year⁻¹ for the deep ocean according to the extended table from Schmidtko et al., 2017.*

The value for the deep Atlantic is already mentioned in the previous lines (line 383-384: "The deep North Atlantic is one of the few ocean regions that has not experienced significant deoxygenation in recent decades (Schmidtko et al., 2017), which may be related to the continued formation and export of LSW and NADW."). However, we do not currently mention the exact value from the Schmidtko et al. paper, and will include this in the new version. Nonetheless, we believe that mentioning the global value is still important for context.

Rewrote this part to include 1.7 Tmol/year number (lines 495-501)

*3) The calculation of the oxygen export at 53°W of 1.35 Tmol and 1.57 Tmol are of course approximations, so I think it should be mentioned. The same for the calculation of the oxygen consumption of 2.2 Tmol by organisms. Regarding this calculation specifically I do have some concern. The authors use a global value of aOUR of 0.1 µmol kg⁻¹ year⁻¹ at 1500 m from Karstensen et al., 2008. First of all, the calculation was made separately for the Pacific and Atlantic oceans, although it seems to be the same for both oceans. Nevertheless, I would indicate the value for the Atlantic. Moreover, the aOUR values are on an order of 10 µmol kg⁻¹ year⁻¹ below the euphotic zone and decay exponentially with a value assumed to be 0.1 µmol kg⁻¹ year⁻¹ at 1500 m. The authors chose this last value for their calculation, however the isopycnal range they considered "27.68 kg m⁻³ and 27.8 kg m⁻³" is not exactly at 1500 m. It is of course changing according to location, so the layer can be as deep as 1500 m but also as shallow as about 200 m or shallower so basically if you always assume that the thickness of this layer is constant (800 m, which is also an approximation because it of course not constant) then your consumption value varies from about 200 Tmol to about 2 Tmol which means the southward export of LSW might supply between 0.71% to 71% of the oxygen demand in this layer.*

We addressed this comment by redoing our calculation using the depth-varying profile from Karstensen et al., see discussion in supplement file. The resulting value is about 3.79Tmol/year, so 42% would be supplied by the LSW export. We will include this as a range in the revision (i.e. 42-71%)

The new calculation is now mentioned in the revised text, and the estimate of O2 supply is given as a range (lines 503-506)

Technical Comments

*Line 31*: *Irminger Water appears here the first time. You could think of define already here the acronym (IW). In the rest of the manuscript, I noticed that sometimes you write Irminger Water other time IW. I think the correct way is once the acronym is defined to stick with that. So please replace Irminger Water with IW in the text after that is defined in line 31. The same for Labrador Sea Water (LSW). Once defined it should be mentioned always as LSW. I found some of them which I highlighted further down but I might have missed some more.*

*Line 31*: *"originating in the Irminger Sea" instead of "originating in the Atlantic Ocean"*

*Line 83*: *wasn't Pickart et al., 1997 the first to define LSW at this boundary. It should be cited before Zantopp et al., 2017.*

*Line 123*: *"LSW" instead of "Labrador Sea Water"*

*Line 124*: *"LSW" instead of "Labrador Sea Water"*

Suggested changes above will all be included

*Line 125 until end of session 2.2*: *here Figure 3 should help to understand the method. But I found this paragraph a bit confusing. Figure 3 is made out of three panels (a, b and c) they should be motioned in the text and help to understand the method.*

We will move part of the caption of this figure to the main text, and rewrite it to make the method more clear

Moved part of caption to main text and rephrased it (lines 165-171)

*Line 140*: *could you give a precise range instead of bigger than 0.5?*

We can add the range if needed; see also discussion below

Range added to text (line 198-199)

*Line 140*: *I do not follow the argument here. If you say that the correlation between $O_2$ saturation and temperature is high it should be an indication that concentration changes are due to temperature-driven solubility differences unless the correlation is lower that the one between $O_2$ and temperature. That's why I suggested above to give a range and not simply saying that is bigger than 0.5.*

We believe that the overall statement is true, but it may have been phrased in a confusing way. "Saturation" in this case refers to saturation percentage, so if changes were purely due to solubility (I.e. saturation stays at a constant percentage, but concentration changes with

temperature), then the correlation of saturation percentage with temperature would be zero, but correlation of O2 concentration and temperature would be high.

We therefore interpret the fact that there is still a high correlation with saturation percentage to show that this is not the case. Another (and perhaps better) way to phrase this is that the two water masses discussed in the paper are distinct in both O2 concentration and O2 saturation percentage, with both being higher for LSW. We will rephrase this paragraph to more clearly state this

We rephrased this paragraph as suggested above to explain the significance of the correlation with saturation percentage (lines 198-204)

*Line 154: February, March and April 2017 or 2018 or both?*

The values given are for 2017, we will add this information

*Line 154 & 155: "oscillate between values typical of the months prior" what do you mean?*

The word "oscillate" may not have been a good choice here, we will change it to "vary between […]"

*Line 179: It looks like you used a dataset that is not described in the data and method session. Please add a description of the dataset from Holte et al., 2017 in the data session.*

We will add a section "additional data" under data and methods where this will be added

Lines 139-141 in newly added section 2.2

*Line 189: "…but are more concentrated towards the sides of the patch…" To me doesn't look like it. A lot of the red dots are in the middle of the patch. What is more concentrated on the sides are the yellow and green. Do you mean that?*

This statement referred to the fact that there are some differences in the distribution between the contours and the points showing convection. For example, there are parts in the interior Labrador Sea where the convective activity inferred from the Holte et al. dataset is maximum, yet there are no points from the float dataset showing convection.

This is likely due to the different methods, with the contours showing a fraction of all profiles measuring convection, while the points only show the last profile with deep mixed layers for each float; so if a float measures several profiles with deep mixed layers in the convective interior

while traveling towards the boundary current, only the last profile would be shown as a "convection profile" in the figure.

We will rewrite this sentence to be less confusing, to "The points generally line up with the mean picture from the Holte et al. (2017) data, with differences between the two likely occurring due to continued modification during convection."

Included the revised sentence above in the text, in line 265-266

*Line 201: "a handful", you could quantify that with a number, 7 right?*

Will include the exact number in the revision

*Line 215: IW instead of Irminger Water*

*Line 215: LSW instead of Labrador Sea Water*

*Line 227: $O_2$ instead of O2 (twise)*

*Line 229: $O_2$ instead of O2*

Changes above will be included

*Line 229: What is the central $O_2$ bin?*

Central O2 bin refers to the most commonly measured concentration for each month

*Line 242: "another." Instead of "another:"*

Will be changed

*Line 246: "Oxygen concentration in the interior before January are lower than those at 53°N during March-July…" to me it doesn't look too much lower. Maybe you could quantify it? Moreover, the SeaCycler mooring has two picks in January. Could you explain why? I think this should be mentioned in the results.*

The value of the central bin (304uM for Seacycler in December, 306uM for K9 in April) is fairly similar, particularly considering measurement uncertainty. However, 304uM is also the maximum value measured at Seacycler in December, while the maximum value at K9 in April is 315uM. So export of older water that is already present in the interior of the basin before

convection starts in January cannot explain the high boundary current O2 during the spring. Some numbers will be added to the text to make this statement more clear.

The two peaks in January occur because convection reaches the depth of the sensor some time during the month, so measurements before show generally lower O2 values, while measurements after show higher ones. We will add this information to the manuscript as well

We moved this paragraph, and added a more thorough explanation of the O2 concentration difference, as well as discussing the two peaks in January, Line 348-353

*Line 273: "LC" instead of "Labrador Current"*

*Line 279: "density." Instead of "density:"*

*Line 362: "salinity." Instead of "salinity:"*

*Line 372: "LSW." Instead of "LSW:"*

Changes above will be included

*In section 4.2 (line 375) you calculated a supply of oxygen for the 6 months (March to August) of 1.35 Tmol from an increase oxygen increase of 6 µmol/L. Could you specify the increase of oxygen for the whole year as well? I guess the 1.57 Tmol/year that you calculated for the whole year is determined by this value since all the other parameters stays the same (mean velocity, layer thickness and section width), otherwise it would be 2.7 Tmol/year.*

The average increase for the whole year is 3.5 µmol/L, this will be added in the revision

*Line 376: "21.2 mol m$^{-2}$" instead of "21.2 mol m$^{2}$"*

*Line 437: "NADW" instead of "North Atlantic Deep Water"*

Changes above will be included

**Figures**

*First of all, I think the authors did a good job with the figures, they are clear, with a good choice of the color schemes and font size. So here are just few suggestions for further improvements:*

**Figure 3:** *Figure caption is really long and the second part it looks like it belongs to the text in the session and not as caption. Moreover 27.74 kg m$^{-3}$ should not be written in italic. Figure 3a it is a bit confusing what is displaying since it is nowhere written what is the thick green line and what are the thin white lines. Figure 3c, why the dark blue line stop at 27.75 kg m$^{-3}$?*

The second part of the caption will be moved to the main text, and we will add clarification to the caption for Figure 3a. In Figure 3c, the dark blue line stops at 27.75 kg/m3 because this is the surface outcrop for this profile, so there is no water at lighter densities.

Moved part of figure caption to main text, added more information about figure 3a, and added explanation about the dark blue line (line 165-171)

**Figure 5**: *The reference at 600 dbar to me seems unnecessary. Moreover, the comparison with the work from Atamanchuk et al., 2020 (purple stars and ellipse) and from Pacini et al., (2020) (green symbols and ellipse) sounds interesting but I do not find it discussed in the paper. If you put that into a figure then it should be discussed, otherwise remove it. Finally, **how is the anomaly calculated?***

The reference to 600dbar will be removed to be more clear. The two references are the studies used for the points shown in green and purple in the figure, which are typical properties of IW and LSW.  The oxygen anomaly is relative to the mean for each measurement location, which will be mentioned in the revised figure caption

**Figure 6**: *The blue to red colormap for absolute values is a bit confusing, especially if it has been used in the previous figure for the anomaly. Also here the reference to 600 dbar is not necessary.*

We will change the colormap and remove the 600dbar reference.

**Figure 7:** *the red dots are too small; I can hardly see them. Could you make the symbol the same size?*

We will increase the symbol size.

**Figure 10**: *The K9 is already displayed in Figure 9a, it is a repetition. Comparison can be easily done by looking at the two figures and best if you plot K7, K8 and K10 one below each other and*

with the same length of K9 in figure 9a. Moreover, I would also put for the 3 other moorings the black line showing the bin with the highest number of observations.

These suggested changes will be included in the revision.

Reviewer 2

Review on "Oxygen export to the deep ocean following Labrador Sea Water formation" by Koelling et al.

*General comments*

Using moored data at the exit of the Labrador Sea (~53°N), Koelling et al. reported the seasonal variability of oxygen concentration in the outflowing boundary current. The origin of the oxygen signal in the boundary current was further discussed by comparing property fields between the boundary current and basin interior, and by tracking spreading pathways of newly-convected Labrador Sea Water (LSW) with Argo floats. Finally, the importance of Labrador Sea convection was emphasized by estimating the oxygen supply from the Labrador Sea to the deep North Atlantic.

I find the manuscript well-written and the focus on oxygen export is of interest to many people in physical oceanography and biogeosciences. Even so, I still have one major suggestion. The Labrador Sea has been extensively studied, both in terms of the boundary current variability and the spreading pathways of LSW. Some of the findings in the current study, such as the seasonality and LSW pathways, are similar to previous work. Besides the oxygen export estimates, what new/different information is revealed by the current study? Specifying the novel findings in a Conclusion paragraph will greatly enhance the importance of this work.

A paragraph was added at the end of the conclusions to summarize the new findings, as well as discussing how our results help connect several aspects of the LSW export process that had only been discussed individually in previous studies (line 571-577)

*Specific comments*

[1]. "The export of oxygen from subpolar gyre is ~71% of the oxygen consumed annually in the upper NADW layer in the Atlantic Ocean between EQ and 50N". This number was only roughly estimated in the Discussion and should not be listed as a key point in the Abstract.

We will remove the exact number and replace it by a more general statement, e.g.: "The export of oxygen was used to estimate the importance of this direct southward pathway of LSW for supplying oxygen that is consumed in the upper North Atlantic Deep Water layer between the equator and 50N"

We left the statement in the abstract, but replaced the 71% number by a range as suggested by reviewer 1 (line 20)

[2]. *Lines 62-64: I believe the importance of LSW in supplying oxygen to global ocean is well known, as stated at the beginning of the Introduction. I suggest the authors to refine their conclusions and implications by specifying the novel findings of the study.*

While it is true that the importance of the Labrador Sea is well known, we are not aware of any previous efforts to directly link deep convection, LSW export, and oxygen supply to the rest of the basin, or to quantify this effect. We will rephrase this paragraph to more clearly highlight the novel aspects of our findings.

Rephrased this paragraph to better highlight the novel aspects of the study (line 77-79)

[3]. *Figure 2 caption: Please check "$\sigma_1$".*

[4]. *Lines 88-89: Please include 27.74 isopycnal in Figure 2.*

[5]. *Line 94: It may not be appropriate to call the saltier waters in the Labrador Current as "remnant of Irminger Water" since the waters are significantly mixed. "Remnant" sounds like that a part of Irminger Water is not mixed but retains in the boundary current. Maybe you can simply say it is a mixture of Irminger Water and convective water.*

Comments 3-5 will be addressed as suggested

[6]. *Line 116: You are defining the export from interior Labrador to the boundary current, instead of the export out of the Labrador Sea (stated in Line 110). Floats that once exported to the boundary current (and stayed for at least 2 cycles) could possibly re-enter the basin interior. A more rigorous selection would be to choose floats that entered the boundary current and stayed there until they exited the Labrador Sea (53N). That would give a smaller number of usable floats but appear to be more appropriate.*

We appreciate the comments on the definition of export, which were also shared by other reviewers. We tested the suggested more rigorous criterion, and found that it did not change our results - see discussion in supplement file for details. We will also change the wording in the method section to clarify that we are defining the input of LSW into the boundary current, rather than the export south of 53N

Changed wording throughout the document to refer to "input of LSW" rather than "export" for the calculations from Argo data (e.g. line 152), in order to stress that the Argo analysis focuses on when floats enter the boundary current, rather than when they leave the basin. Added reference to stricter criterion (line 154-156)

*[7]. Figure 3 caption: Suggest using "convective interior" or "interior" instead of "convection".*

Will be changed as suggested

*[8]. Figure 4: Please make y-axis consistent between panels. The increase of oxygen in March 2018 is quite sharp compared to March 2017. Could you please include a few comments on that?*

The y-axis is the same for K8-K10, and is only shifted down by 5uM for K7 since mean concentrations are lower, but the vertical extent of each axis is the same. This is done to "zoom in" as much as possible on the curves to highlight the variability while still showing all data points. We will add this information in the caption, but we can also adjust the axes limits to be consistent between all panels if preferred.

Reviewer 4 suggested to add a discussion of interannual variability, which will be included in section 4.2. We can add a brief analysis of the differences between 2017 and 2018 in that context.

Added text to figure caption to mention different axis ranges. Discussion of differences between 2017 and 2018 was added to section 4.3, line 517-520

*[9]. Figure 5: Why are some high oxygen profiles (~10 μmolL⁺) associated with relatively high temperature (~3.5C) and salinity (~34.9) at K10? Does that indicate a different source/route of the oxygen input?*

This is an interesting question, but probably goes beyond what is discussed in this study. Note also that these are just two days out of the entire record. These measurements also occur at a time with the highest velocities measured at K10 (>25 cm/s southward) and show higher densities than usually measured at this site, so they may be related to mooring knockdown to depths much greater than 600m.

*[10]. Lines 175-176: If convection occurs in the boundary upstream of 53N, wouldn't the density anomalies propagate downstream to the array? Do you see that density signal at 53N?*

There is no clear density signal associated with the arrival of newly convected water, as discussed in line 167. Whether or not a density anomaly associated with upstream convection would propagate to the array would depend on the degree of mixing and restratification occurring between the two points. The lack of a detectable density anomaly could therefore mean that these processes have obfuscated the signal of convection by the time the water reaches 53N

*[11]. Line 211: I do not think everyone is familiar with spiciness. I suggest including a paragraph describing how the variable is calculated.*

We will include a short description of spiciness.

Description included in line 290-291

**[12]. Lines 238-240 & Figure 9b:** *If you separate the exported LSW from convection in the boundary current, will the temporal variability of LSW input be different?*

This is a very interesting question, which we discussed in the supplement file with a new figure showing the input from boundary and interior separately. There does appear to be a difference in timing between boundary and interior convection, and we propose adding a brief discussion of this in the revised manuscript

We added information to figure 9b showing separately the input from boundary and interior convection. A brief discussion was added in line 341-347

**[13]. Lines 246-248:** *I do not think I fully understand this statement. Could you elaborate?*

This statement is meant to show that the increase in O2 at K9 cannot be related to increased export of older LSW; i.e. the O2 in the boundary current increases because of recently ventilated water. Since almost all measurements at the Seacycler mooring in December lie in the 304uM bin (and none are above it), they cannot be the same water mass found in the boundary current in February-April, with concentrations as high as 315uM.

This statement was rephrased in the text to be more easily understood, Line 351-353

**[14]. Line 252:** *The 12-month time series of LSW input is heavily smoothed (5 months). The 1month lag could be entirely artificial.*

The time series is smoothed with a 5-point (20-day) filter, not 5 months. We also show the data in the original 5-day bins in the supplement file, and can add this to the revised document if needed to show the actual data along with the smoothed curve.

Data in 5-day bins were added to fig 9b

**[15]. Line 266:** *What is "the boundary with of the LC"?*

**Line 273:** *What is "the border of the LC and DWBC"? I suggest showing the velocity structure superimposed with the mooring locations for better illustration.*

The velocity structure is shown and discussed in depth in a recent paper on the 53N current meter data, Zantopp et al. (2017). We will add a sentence in the revised manuscript referring the reader to that paper for more details

*[16]. Lines 290-291: This contradicts Figure 2, which shows smaller oxygen concentrations at K10 compared to the other moorings.*

We are referring here to the spread of O2 measurements shown in figure 10. As seen in figure 2, the K10 mooring is near a larger horizontal oxygen gradient, so the spread may be related to meanders of the current, as mentioned in the following lines.

*[17]. Line 349: I do not think ISIW, the locally convected water in the Irminger Sea, is the same as Irminger Water. The IW is the saltier and warmer Atlantic-originated water that flows in the boundary current. What enters the Labrador Sea is likely a mixture of ISIW and IW.*

We appreciate the clarification, and will rephrase this in the revision

*[18]. Lines 377-378: I think this is a very intriguing estimate. If recirculation at K10 is considered, how will the (net) export percentage change?*

*[19]. Line 434: Again, recirculation is not considered and the percentage (71%) could be an overestimation.*

This is again an interesting question, but may be beyond the scope of the current study to address. Zantopp et al. (2017) quantified the recirculation strength from LADCP data, and found that it is typically about 10-20% of the DWBCs strength. However, the effect of the recirculation on the net oxygen export will also depend on how much of the newly convected LSW enters the recirculation, which may also have some input of water from the North Atlantic Current. We can add the 10-20% number to the manuscript as an upper bound for how much LSW could be recirculated northward

The potential effect of recirculation on oxygen export is briefly discussed in line 491-493

Reviewer 3

*Minor Revisions*

1. *More appropriate handling of uncertainty in the back-of-the-envelope calculation.*

*I commend the authors for attempting to put numbers to the supply of oxygen to the deep waters of the Atlantic - this is a valuable contribution. However, I felt that numbers like $1.57 \times 10^{12}$ mol O2 yr$^{-1}$ and 71% imply a level of accuracy that is inconsistent with the uncertainty and*

*assumptions that have gone into their calculation. I would ask that the authors present the numbers as a range that takes into account the uncertainty associated with each component of the calculation.*

*I note that Dr. Stendardo also picked up on this point, providing further specifics on one potentially large source of uncertainty.*

We discussed possible changes that to include uncertainty estimates in the supplement file. The proposed changes would allow us to add an uncertainty estimate to the O2 export (e.g. $(1.57\pm 0.42) \times 10^{12}$ mol O2 yr$^{-1}$), and give the estimate of O2 consumption in the North Atlantic south of 50N as a range (e.g. 42-71%).

We used LSW transport uncertainty from Zantopp et al. to get an uncertainty for the O2 export (Lines 478-481), and added a different method for estimating the O2 consumption, listing a range for the supply from LSW (line 503-506)

1. *Inference of timescales and hypothesis of eddy-driven exchange from central Labrador Sea.*

*The authors use the difference in the timing of the seasonal oxygen peak in the central Labrador Sea and at the moorings to infer a speed associated with oxygen transport between the two (paragraph beginning Line 255; discussed again Lines 320-323). Noting that this is much larger than the speed of the time-mean flow, they use this is as evidence for the role of time-varying, eddying flow in driving this transport. However, the authors previously argued convincingly that the boundary current peak was more than likely arising from convective processes within or close to the boundary current itself. Therefore, inferring instead a timescale of exchange from the central region appears inconsistent with this. Please could the authors clarify if I am misunderstanding something here, or else revise these statements, which I don't believe their observations support.*

The previous discussion suggested that the initial increase of oxygen (i.e. February-March) at 53N is associated with boundary convection, but both boundary and interior convection contribute to the increase later in the season. This paragraph was meant to only discuss the mechanism by which the part of LSW convected in the interior enters the boundary current. We concede that it was unfortunate to phrase it as "If the bulk of the LSW arriving at 53° N originates in the center of the basin near the SeaCycler mooring", which might be interpreted to imply that convection in the boundary current is not important; we will rephrase this paragraph to clarify.

The importance of both interior and boundary convection is also evident from figure 7, and the new version of figure 9 shown in the supplement file. The new analysis also shows that the peak

export does occur somewhat later for the interior LSW than for the combined estimate, so the time scale for export of LSW sourced from the interior will have to be revised to 1-2 months, implying a range of export speeds from 4.4-9 cm/s.

The revised version of figure 9b shows that in fact the time difference between Seacycler convection and input of interior LSW is one month, so we kept the 9cm/s number. The paragraph was rephrased to specifically focus on input of LSW from the interior (line 354-359)

1. *Evidence for and against local ventilation at 53N*

*The authors make an effort to affirm that the oxygen and watermass changes at the mooring locations are most likely driven by processes taking place upstream, rather than occurring locally (i.e. from surface forcing impacting the water column above). While I agree that this is probably true, I thought some of the lines of evidence presented were not entirely conclusive. In particular, the authors cite the lack of density changes over a seasonal cycle (paragraph starting Line 67). However, Fig 6 (and to some extent Fig 5) do indeed show density changes on the order of 0.025 kg m-3 over the seasonal cycle, indicative of a slight warming and freshening concurrent with oxygen increases. These density changes could be significant in this weakly stratified region. Of course, such density changes may or may not be indicative of local surface forcing (more likely there is a seasonality in the doming of isopycnals coincident with the strength of the gyre) but the authors' assertion that there are no changes is likely to confuse readers.*

*The authors state that an absence of density changes confirms that local ventilation is not taking place. However, convection (and therefore local ventilation) to a certain depth, need not be accompanied by diabatic transformation at that depth. It is possible that homogenization of the upper water column could take place without changing the density at 600m itself, since it requires the densification only of the waters above. Diabatic changes at 600m would indicate a mixed layer extending into stratified water much deeper than 600m.*

*It may be the case that the strongest evidence for the absence of local ventilation comes from the lack of static instability relative to surface density, which the authors allude to on Line 170, but the data for which they don't show. The authors should consider showing that data, and centering this argument in their reasoning for changes being driven by upstream processes.*

We appreciate the suggestions and explanation, and will focus the argument more on the near-surface density data in the revision.

A figure was added showing the density at K8 (figure 6b), and the discussion was changed to focus on this part of the argument (lines 236-246)

1. *The role of solubility in oxygen variations.*

*Lines 139-141: I didn't follow the argument concerning the correlation of oxygen saturation with temperature, and how this refutes the possibility that oxygen concentration variability simply reflects solubility changes. Further, I was not sure why solubility-driven changes should be considered less relevant here? I would have, perhaps naively, thought that solubility derived changes would be a relevant and important mechanism by which LSW is oxygen replete relative to warmer waters. Could the authors please elaborate on their explanation here and clarify the point that I am missing?*

(copied from response to reviewer 1) We believe that the overall statement is true, but it may have been phrased in a confusing way. "Saturation" in this case refers to saturation percentage, so if changes were purely due to solubility (I.e. saturation stays at a constant percentage, but concentration changes with temperature), then the correlation of saturation percentage with temperature would be zero, but correlation of $O_2$ concentration and temperature would be high.

We therefore interpret the fact that there is still a high correlation with saturation percentage to show that this is not the case. Another (and perhaps better) way to phrase this is that the two water masses discussed in the paper are distinct in both $O_2$ concentration and $O_2$ saturation percentage, with both being higher for LSW. We will rephrase this paragraph to more clearly state this

We rephrased this paragraph as suggested above to explain the significance of the correlation with saturation percentage (lines 199-204)

1. *Clarity in Figure 7.*

*I found myself confused about the labelling of points in Fig. 7. I understand from reading the caption that all of these points are convection locations, with the yellow and green points being the convection locations of floats that subsequently showed export across 3000m. However, the marker styles and legend could be read as suggesting that convection is taking place only at the red points, and that the yellow/green points are perhaps the locations of export or some other notion associated with the pathways. Either way, I initially inferred that the yellow, and green points, are somehow functionally distinct from the red points. In fact it is the case that all of the yellow points are also red points, and all of the green points are also red and yellow points. A possible solution would be to keep all the marker styles the same, but with different colors - clarifying that these points fundamentally show the same thing (convection location) but with distinguishing characteristics (pathways following convection). Likewise, the wording of the legend should be changed to be clearer in this regard (e.g. "convection profiles, float not exported; convection profiles, float exported (any time); convection profiles, float exported Jan-Feb").*

Thank you for the suggestions, we will change the figure accordingly to be easier to understand

**Reviewer 4**

*General comments:*

1. *Convection in the Labrador Sea exhibits pronounced interannual and decadal variability (e.g. Fisher et al. 2010, GRL). For example, convection was particularly intense between 1987-1994 (Marshall et al. 2001, J. Clim.). All of the results in this paper are derived from a single year of measurements, but no discussion is given to how representative these measurements and findings are of other years, and of the long-term mean behavior of the Labrador Sea. The authors should discuss these issues clearly throughout the abstract and manuscript. It seems to me that the results might differ substantially if the measurements were made in a year of particularly intense convection in the LS; if this is the case, the authors should clarify throughout the manuscript that their results apply to the particular year of measurements, and that its generalizability to other years remains unclear.*

The effect of interannual variability in Labrador Sea Water formation is an important question that we unfortunately cannot adequately answer with the current data set, but we hope to address it in a future study using a longer record.

The overall outflow in the Labrador Sea Water density layer has been shown in previous studies to not have a clear connection to the strength of convection, as shown in the Fischer et al paper you mentioned: "[…] the annual mean flow varies at all depth levels by the order of 10% relative to the 'decadal' mean, and again there are no detectable systematic trends in the boundary current intensity.", as well as other studies using 53N data (Dengler et al., 2006, GRL).

Nonetheless, the amount of LSW entering the boundary current from the interior, and/or the strength of convection in the boundary current itself, will likely be different between period like the 1987-1994 convection regime compared to periods like 2005-2008 when convection was shallower. Accordingly, the properties of the outflow may be closer to the IW endmember as used in our study during a weak convection period, and closer to "pure" LSW in a strong convection period, which would also affect the oxygen content of the outflow. We will add a paragraph to the discussion in section 4.2 to address the subject of interannual variability, and discuss how it might change the results.

Added a paragraph at beginning of section 4.3 discussing results from previous studies on interannual variability in LSW properties in the interior, and LSW transport out of the basin (line 517-528)

1. *The authors measurements exhibit two key features that mark the arrival of LSW at the boundary current array: (i) a rapid shift of the water mass properties toward lower spice and higher O2, and (ii) increased temporal variability in O2 (and presumably spice) about*

*the monthly mean. While the authors discuss the drivers of feature (i) extensively and conclusively, I could not find any explanation for feature (ii). Arguably feature (i) is more directly relevant to the rates of LSW and O2 export via the boundary current, but the processes underlying (ii) should at least be discussed, even if only to offer some speculation as to its origins based on previous studies.*

We do mention a possible explanation for the high temporal variability in section 3.1, lines 164-166: "This stark heterogeneity of properties may suggest that newly formed LSW is rapidly exported out of the Labrador Sea in February, and some of it is transported with the boundary current without much mixing with the surrounding water." and in the conclusions, lines 423-425: "LSW first arrives at the offshore moorings in the second half of February, with a wide range of temperature, salinity, and oxygen properties reflecting a sporadic input of heterogeneous LSW formed in or near the boundary current."

However, we concede that the text currently does not give a sufficiently detailed explanation of the possible drivers. We believe that the large range of properties is due to spatial inhomogeneities in the boundary current during the early stages of convection; i.e. convection occurs in "patches", and at the start of the convective season, newly convected water can coexist next to old boundary current water with properties closer to IW, and/or convected water with slightly different properties. As convection progresses, the boundary current becomes more and more homogenized, and the high-frequency variability subsides. High-frequency variability in properties in the earlier stages of convection is also observed in Cuny et al. (2005, JPO), who discuss convection within the boundary current further upstream. They also relate this difference to spatial inhomogeneities, which they suggest could result from spatial variability within the boundary current regime in atmospheric forcing or preconditioning.

We will add a brief discussion of these possible explanations in section 3.2 in order to more clearly state what the drivers for the high-frequency variability could be.

Added discussion of Cuny et al. paper in section 3.1 (line 228-235), and also referenced this in section 3.2 (line 302-305)

*Specific comments:*

**Abstract:** *The abstract is accurate, but I was surprised to find no mention of the Argo float analysis, nor the authors' inferences regarding the supply of LSW to the boundary current.*

Thank you for the suggestion, a sentence about the Argo analysis will be added.

Rephrased abstract to explicitly mention the Argo data, and how they support the picture of LSW export from the moorings, line 8-12

*L19-22: These claims should be supported by citations.*

Citations will be added

*L71-72: Please clarify (briefly) in what way the observations are optimized. I presume the authors are referring to the selection of instrumentation and the locations of the instruments across the section*

The observations are optimized primarily to capture the variability of the boundary current transport, as well as the depth of interfaces between water masses. We will add this information to the paper

*Fig. 1: This is an excellent introductory figure. However, I did not see any details given in the text regarding the calculation of the mean salinity. E.g. how are the Argo floats binned into horizontal grid boxes to create this figure? Is this an average over all seasons? What criteria (e.g. quality controls) are applied to decide which Argo profiles to include/exclude?*

The Figure is an average over all seasons from 20 years of data, using all data that were flagged as 'good' in the Argo catalog. The data are binned into overlapping 0.25x0.25 degree bins. The caption will be updated to include this information

Additional information was added in fig 1 caption, and in section 2.2 (lines 134-140)

*One aspect of the Labrador Sea circulation that this figure does not highlight is the properties and volume of LSW and other water masses. A section across the central LS would show this nicely (though Argo measurements may be too shallow), and would complement the discussion in section 1 (e.g. lines 30-33, lines 47-51). Note that this simply a suggestion for the authors, which they are welcome to take or leave as they see fit.*

We will add more specific references to refer the reader to existing section plots of the Labrador Sea; i.e. Yashayaev et al. (2017, GRL) for T, S, and O2 sections along AR7W, and Zou et al. (2020, Nat. Geosciences) for average T and S properties from the OSNAP array.

Added a brief statement about LSW and IW properties in hydrography with references to Yashayaev & Loder and Zou et al papers, Lines 56-58

*Also, why have the authors used Smith and Sandwell (1997), rather than more recent bathymetric products?*

The product used is the most recent version of the Smith and Sandwell product (SRTM15+ from 2019), but we used the original reference for the dataset. We will update with a citation of the actual dataset used.

Citation added in section 2.2, Line 143-144

*Finally, the directions of the arrows in this figure are difficult to discern. I think I see a flow reversal across the mooring array, but it is difficult to tell. I suggest that the authors use larger, wider arrowheads here.*

The figure will be changed with larger flow vectors

*Fig. 2, L77-79: I understand that the plotted properties are averages across four cruises. Were the measurements made at the same locations on each cruise? If, so it would be appropriate to indicate these locations on the plot. If not, then some additional explanation is required to explain the procedure via which the measurements were gridded to create these plots.*

The measurements are mostly from standard stations occupied during each cruise. **We will add** symbols showing the locations from one representative cruise.

Symbols added to fig. 2, description of CTD data in Line 129-133

*L113-115: The authors should explain their choice of density threshold for the mixed layer depth. If this choice is standard then citations should be given, or if they have selected it then they should explain why they used this specific threshold, and discuss the sensitivity of their results to this threshold.*

References will be added.

*L116-118: I did not see a similar export criterion for determining Lagrangian floats in the cited studies. By the authors' definition, floats will be considered to have been "exported" if they merely enter the boundary current across the 3000m isobath, remain there for two subsequent profiles, and then leave the boundary current without returning. This does not conform to my conception of "export", and requires further explanation or possibly modification.*

The export criterion is similar to definitions used in the Georgiou et al. study in the sense that they tracked floats that were found in the boundary current near Greenland and determined whether or not they stayed in the boundary current or entered the interior, noting that floats

sometimes have short "excursions" into the interior before returning to the boundary current. The criterion was meant to allow for such short excursions after floats enter the BC.

We understand the concern about the definition of export, and addressed this with a short discussion and figures in the attached supplement. We tested a stricter criterion, requiring floats to leave the Labrador Sea in the boundary current south of 53N, as suggested by another reviewer, and found that the resulting export estimate is almost unchanged. We hope that this is sufficient to show that our definition does not lead to major biases. We will also change the wording in the methods section to clarify that the method looks at floats entering the boundary current, rather than those that are exported south of 53N, e.g. by calling it "input into the boundary current" (as done in Figure 9b) rather than "export"

Changed wording throughout the document to refer to "input of LSW" rather than "export" for the calculations from Argo data. Added reference to a stricter criterion, which would not change the results presented (line 154-156)

*Fig. 3:* The second paragraph of this caption really belongs somewhere in the main text.

Paragraph will be moved to main text

*Table 1 captions:* "locations", "depths" and "drifts" should be plural in this caption.

Caption will be changed as suggested

*Fig 5.:* A very large number of data points are shown on these diagrams (over 70,000 at each morning, assuming that the 15 minute-frequency data are used). Consequently, many of the points overlap, obscuring a substantial fraction of the oxygen measurements. To clarify the presentation I recommend instead binning the oxygen measurements into discrete T/S bins, and then plotting the mean O2 in each bin (although other statistics, such as the standard deviation, may be of interest too) on a regular T/S grid.

Thank you for the comments. We already accounted for this, but unfortunately did not mention it in the figure caption. The data points shown in figure 5 are daily averages, and we will add this information to the caption.

*L163-164:* The implication here is that the water properties vary much more slowly over the rest of the year, but the authors have not plotted the time series that would show this. It would help to show plots analogous to those shown in Fig. 4, but for T and S rather than O2.

While this would be instructive, we feel that the T and S variability is not sufficiently discussed in the paper to justify adding two more figures. Instead, much of this information can be gleaned from comparing figures 5 and 6, and we propose to add a statement to that effect in the text, e.g. "[…] observed within just 20 days (see Figs. 5 and 6)"

*L171*: Citations are required to support the claim that this criterion is "commonly used".

Citation will be added

*L187:* Here and elsewhere in the manuscript, the authors should be clear that they are specifically identifying occurrences of what I would refer to as "deep convection". More generally, convection takes place frequently in the surface mixed layer due to local static instabilities, but only penetrates deep enough to form LSW in the interior of the Labrador Sea during winter.

We will clarify this by calling it "deep convection" throughout the manuscript

We felt that it was better for the flow of the text to keep the term "convection", so instead we clarified in the introduction that we use "convection" to refer to deep convection (lines 31-34)

*L190-191:* It may be that I am misunderstanding this statement, but it looks to me like most of the floats measuring deep convection are offshore of the 3000m isobath.

This is phrased somewhat confusingly, and will be clarified in the revision. The statement is meant to say that out of those floats that do measure convection inshore of the 3000m isobath, most are found close to the interior convection patch, i.e. concentrated on the western side of the basin at about 57N.

*Figs. 8-9 and in the text:* Spiciness is missing units; I believe they are usually kg/m^3. If it has been normalized then the normalization should be given.

Units will be added

*L228-229:* I found "a wider range of export and mixing time scales" to be unclear, and I do not think that the authors have provided evidence to support this claim.

This statement was meant to convey that the boundary current upstream has received more LSW input from both the interior and local convection, and this leads to a more homogenized set of properties. We will change this sentence to reflect this – e.g. "By May, the boundary current has become more homogenized, as evidenced by the higher number of observations in the central O2 bin, suggesting an increase in LSW input."

Changed to "By May, the boundary current has become more homogenized, as evidenced by the higher number of observations in the central O2 bin.", line 308-310

*L237-238:* While the authors clearly explained in section 2 how they identify Argo floats moving from the interior LSW to the boundary current, I am unclear on how they have converted this information into an estimate of the LSW flux into the boundary current. Is this derived from

*some combination the number of Argo floats and the layer thicknesses measured by each Argo flat as they enter the boundary current?*

As explained in the discussion in the supplement file, the "LSW input" estimate is not meant to represent an actual volume flux, but is simply an estimate based on the number of floats entering the boundary current during each time step. We will add this information to the manuscript in the methods section

Added information about the calculation in section 2.3 (line 182-188, eq. 1)

*Additionally, the authors should discuss whether sampling biases may be influencing this calculation. The implicit assumption underlying this calculation is (presumably) that the LSW is densely sampled by Argo floats with similar numbers of samples in each 5-day period. Deviations from this ideal (which seems likely, given the limited number of float locations shown in Fig.7) may introduce biases/uncertainties into the distribution of LSW inflow to the boundary current as a function of time, which should be handled appropriately.*

As touched upon in section 3., Argo floats are a valuable tool, but due to their lagrangian nature are also unable to sample a region with truly homogeneous spatial coverage. We are aware that an estimate of LSW input over time based on a limited number of floats may not fully reflect the true underlying variability, and we will add a sentence to the revision more clearly stating that this may bias the curve shown in figure 9b.

Explicitly mentioned possible biases in the text, line 322-327

*L257-258: Alternatively (as the authors have indicated before) the LSW could originate from convection occurring in or close to the boundary current, as suggested by Fig. 7.*

This paragraph was only referring to the input of LSW from the interior, we will clarify this in the revision

Rephrased this paragraph to specifically discuss LSW input from interior, and referenced new information shown in figure 9b, line 354-359

*L293-294: Should we expect a correlation with the current speed? For advection of O2 down a mean O2 gradient, southward velocities would produce a positive O2 tendency (d(O2)/dt>0), while northward velocities would produce a negative O2 tendency (d(O2)/dt<0). However, the O2 concentration is equal to the time-integral of its time-tendency, so for a fluctuating flow we might expect a stronger correlation between the O2 concentration and the time-integral of the southward velocity than with the southward velocity itself.*

Thank you for pointing this out, we will adjust the argument to no longer use the correlation of O2 and current speed, and instead make a more qualitative statement.

Changed to a qualitative statement, line 395-398

*Also, it looks to me like the modal O2 concentration is higher at K10 than at any of the other moorings (compare the May O2 concentration of almost 310 umol/L with those at the other moorings, for which the modal concentrations only reach ~305 umol/L). If the O2 concentration at K10 is the result of southward advection in the boundary current followed by northward recirculation, how does it achieve higher O2 concentrations than the moorings within the boundary current?*

As mentioned in the text, the K10 mooring can at different times be either within the DWBC or in the recirculation regime, so the highest O2 concentrations don't have to be associated with the recirculation. Also, T/S values for the highest O2 measurements at K10 are close to properties of LSW in the interior of the basin (see fig. 5), so these high O2 values may result from LSW entering the boundary current from the interior and mixing less with IW than further onshore (i.e. at K9).

**Reviewer 5**

*Minor comment*

*Since the authors discuss the differences between the properties of the Labrador Sea Water (LSW) and the Irminger Water (IW) I would expect a comparison of the temperature and salinity (and maybe oxygen?) ranges of these two water masses with previous studies. Last, it is not clear until almost the end of the manuscript that the authors refer to the water mass that is formed during convection in the Irminger Sea as Irminger Water.*

The temperature and salinity ranges from previous studies are what is used in Figure 5 to define the IW and LSW properties. We will also add a reference to Yashayaev & Loder (2016, JGR: Oceans) to refer the reader to section plots of T, S, and O2 to help with interpretation.

*Specific comments*

*Line 61: (and elsewhere) change analyse to analyze.*

*Figure 3: The second paragraph within the caption should be moved to the main text.*

*Line 114: Please add a reference for the definition used to calculate the MLD.*

Suggested changes above will be included

*Line 116: The definition of the export of a float is not very clear. Do you define as export when a float crosses the 3000 m isobath and then reaches to a certain location to be considered out of the Labrador Sea or when a float enters and remains into the boundary current?*

Export is defined when a float enters and remains in the boundary current, but using a stricter criterion such as requiring export past 53N does not change the results (see supplement file)

Rephrased throughout the document from "export" to "LSW input into boundary current". Reference to stricter export criterion added in line 154-156

*Figure 4: Would it be better to have all the stations in one plot with different line colors? Otherwise, please keep the same limits on y-axis for each panel.*

(Copied from response to Reviewer 2) The y-axis is the same for K8-K10, and is only shifted down by 5uM for K7 since mean concentrations are lower, but the vertical extent of each axis is the same. This is done to "zoom in" as much as possible on the curves to highlight the variability while still showing all data points. We will add this information in the caption, but we can also adjust the axes limits to be consistent between all panels if preferred.

Added in figure caption

*Figure 7: Do all these floats reach the 53◦W section after subduction?*

No, these are just the floats that are exported by entering the boundary current; about half of them later reach 53N. However, the timing of floats entering the boundary current does not change if we only consider those that are later exported south of 53N - see supplement file for more details

*Line 201: Please specify the number of floats instead of mentioning "handful".*

Exact number will be added

*Line 235 and Figure 9: I understand that the LSW input into the boundary current is defined based on the float data introduced in section 3.1, but could you specify a bit more the calculation of the LSW input?*

LSW input is simply based on the number of floats exported (as defined above), see supplement file for more details. A more thorough explanation will be added to the methods section

Details on the calculation of LSW input rate were added in line 182-188 and eq. 1

*Figure 8:* Is the separation between LSW and IW in terms of spiciness arbitrary or is it based to previous studies?

The values chosen for spiciness are arbitrary, but the difference in T and S properties are well established from previous studies (i..e Pacini et al., 2020)

*Line 348-349:* I believe that this explanation should be also mentioned earlier in the text.

As pointed out by Reviewer 2, our statement regarding ISIW is actually incorrect; ISIW is not the same as IW, but rather is mixed into IW in the boundary current of the Irminger Sea. We will change the statement to reflect this.